# Early-adulthood spike in protein translation drives aging via juvenile hormone/germline signaling

Harper S. Kim[1,2,3,4], Danitra J. Parker[1,5], Madison M. Hardiman[1], Erin Munkácsy[3], Nisi Jiang [3], Aric N. Rogers[6], Yidong Bai[3,7], Colin Brent[8], James A. Mobley[9], Steven N. Austad [10,11] & Andrew M. Pickering [1,3,5,12] ✉

Protein translation (PT) declines with age in invertebrates, rodents, and humans. It has been assumed that elevated PT at young ages is beneficial to health and PT ends up dropping as a passive byproduct of aging. In *Drosophila*, we show that a transient elevation in PT during early-adulthood exerts long-lasting *negative* impacts on aging trajectories and proteostasis in later-life. Blocking the early-life PT elevation robustly improves life-/health-span and prevents age-related protein aggregation, whereas transiently inducing an early-life PT surge in long-lived fly strains abolishes their longevity/proteostasis benefits. The early-life PT elevation triggers proteostatic dysfunction, silences stress responses, and drives age-related functional decline via juvenile hormone-lipid transfer protein axis and germline signaling. Our findings suggest that PT is *adaptively* suppressed after early-adulthood, alleviating later-life proteostatic burden, slowing down age-related functional decline, and improving lifespan. Our work provides a theoretical framework for understanding how lifetime PT dynamics shape future aging trajectories.

Protein translation (PT) is an essential cellular process playing key roles in growth and development. PT occurs at a high level at young ages, but it then declines precipitously, remaining low throughout middle-old ages in multiple animal species, including humans[1–6]. One might expect that lowering PT would be detrimental to health because it might lead to a shortage of critical cellular proteins and slower protein turnover, allowing more protein damage to accumulate. However, life-long reduction in PT has been reported to slow down aging-related functional declines, prolong lifespan[7–10], and ameliorate cellular senescence and age-related diseases[11–16]. We note that, across animal species, PT is elevated in *early*-adulthood[1–6], implying that life-long PT suppression would impact this critical time window the greatest to promote longevity. Yet, it has been generally thought that high PT at young ages is beneficial to health while PT ends up dropping over time as a passive byproduct of aging. Whether this holds true and how dynamic fluctuations in PT over time impact aging remain unknown. We thus transiently modified PT at different life stages in *Drosophila* and investigated how these modifications impacted aging phenotypes.

[1]Center for Neurodegeneration and Experimental Therapeutics, Department of Neurology, University of Alabama at Birmingham, Birmingham, AL 35294, USA. [2]Medical Scientist Training Program, University of Alabama at Birmingham, Birmingham, AL 35294, USA. [3]Barshop Institute for Longevity and Aging Studies, University of Texas Health San Antonio, San Antonio, TX 78229, USA. [4]Medical Scientist Training Program, University of Texas Health San Antonio, San Antonio, TX 78229, USA. [5]Department of Integrative Biology and Pharmacology, McGovern Medical School at UTHealth, Houston, TX 77030, USA. [6]MDI Biological Laboratory, Bar Harbor, ME 04672, USA. [7]Department of Cell Systems and Anatomy, University of Texas Health San Antonio, San Antonio, TX 78229, USA. [8]USDA-ARS Arid Land Agricultural Research Center, Maricopa, AZ 85138, USA. [9]Department of Anesthesiology and Perioperative Medicine, University of Alabama at Birmingham, Birmingham, AL 35249, USA. [10]Department of Biology, University of Alabama at Birmingham, Birmingham, AL 35294, USA. [11]Nathan Shock Center, University of Alabama at Birmingham, Birmingham, AL 35294, USA. [12]Department of Molecular Medicine, University of Texas Health San Antonio, San Antonio, TX 78229, USA. ✉e-mail: Andrew.M.Pickering@uth.tmc.edu

We find that the transient PT elevation in early-adulthood disrupts protein homeostasis (proteostasis) late in life, triggers age-related protein aggregation, and drives age-related declines. Our findings also indicate that the rapid drop in PT after early-adulthood is critical to reducing proteostatic burden, slowing down age-related decline, and improving life-/health-span. This suggests that age-related decline in PT, instead of simply being a passive byproduct of aging, may help promote healthy aging. Our work provides a theoretical framework for understanding how lifetime PT dynamics shape future aging trajectories and proteostasis network.

## Results

### Blocking the transient PT elevation during early-adulthood improves life-/health-span

In both sexes of *Drosophila melanogaster*, we observed a PT elevation during early-adulthood (Fig. 1a female, Supplementary Fig. 1a male). PT

increased ~5-fold from day 0 to day 2, declining markedly thereafter. This observation was consistent using three independent methods: $^{35}$S-methione incorporation (Fig. 1a, left), puromycin incorporation (Fig. 1a, middle), and in vitro luciferase mRNA reporter assay (Fig. 1a, right). The latter assay measures the capacity of lysates to translate introduced luciferase mRNA[17]. The luciferase assay was thus used to verify that low PT at day 0 was not an artifact of feeding labeled substrates to newly eclosed flies that may use larval acquired amino acids for PT. To investigate the impact of early-life PT rise on aging phenotypes, we transiently repressed PT in early-adulthood (day 0–10) with a widely-used PT inhibitor (cycloheximide, CHX) (Fig. 1b). We also fed flies CHX during late-adulthood (day 40–50) and the entire adult life (Fig. 1c). Consistent with previous studies, CHX treatment throughout the entire adult life significantly extended lifespan. Surprisingly, CHX treatment for only the first 10 days of adult life produced a similar lifespan extension (Fig. 1d). CHX treatment late in life, however, did not

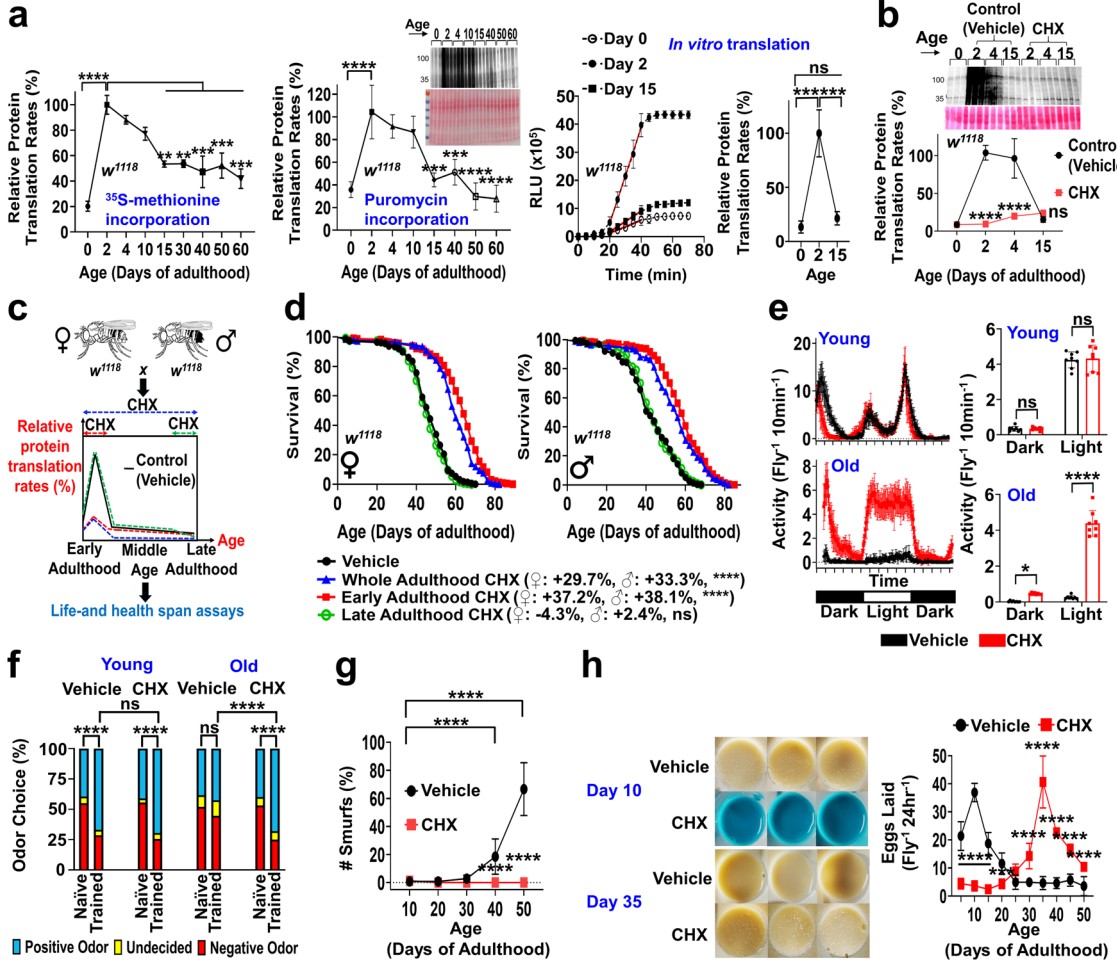

**Fig. 1 | Blocking the transient PT elevation during early-adulthood improves lifespan and healthspan. a** PT across ages, determined by (left) $^{35}$S-methionine incorporation normalized to protein content (n = 3) and (middle) puromycin incorporation normalized to Ponceau staining (n = 3). Analyses compare PT relative to day 2; one-way ANOVA with Dunnett's post-hoc test. (Right) in vitro PT assay with luciferase mRNA reporter and fly extracts. Slope of linear part of luciferase activity curve is used to calculate the rate of PT. n = 12/group; one-way ANOVA with Tukey's post-hoc test. **b** Early-adulthood PT after ±1 μM CHX (day 0–10), determined by puromycin incorporation normalized to Ponceau staining. n = 3/group; two-way ANOVA with Sidak correction. **c** Experimental scheme to transiently manipulate PT in different life stages; 1 μM cyclohexamide (CHX) given to $w^{1118}$ flies during early-adulthood (day 0–10), late-adulthood (day 40–50), or whole-adulthood. **d** Early-adulthood (day 0–10) CHX prolongs lifespan just like whole adulthood CHX. Late-adulthood (day 40–50) CHX does not alter lifespan. Each sex: n = 250/group; log-

rank test. **e** Early-adulthood (day 0–10) CHX reduces age-related deficits in spontaneous activity. Young=day 12, old=day 50. n = 200/group; two-way ANOVA with Sidak correction. **f** Early-adulthood (day 0–10) CHX reduces age-related cognitive deficits in olfaction aversion training. Young=day 12, old=day 50. n = 200/group; Chi-square test. **g** Early-adulthood (day 0–10) CHX prevents age-related deficits in gut-barrier integrity in Smurf assays. n = 250/group; two-way ANOVA with Sidak correction. **h** (Left) representative images of eggs laid on vials after CHX treatments during early-adulthood (day 0–10). (Right) Early-adulthood CHX impairs egg production at young ages, delays fertility peak, and improves egg production at old ages. n = 100/group; two-way ANOVA with Sidak correction. Data shown as mean ± SD. female data for PT assays and healthspan assays shown here. Male data in Supplementary Fig. 1. *p < 0.05, **p < 0.01, ***p < 0.001, ****p < 0.0001. Source data are provided as a Source Data file.

extend lifespan (Fig. 1d, Supplementary Fig. 1b). These results suggest that the transient PT elevation in early-adulthood may be an important driver of aging.

Healthspan was also robustly improved after preventing the early-adulthood PT elevation. For both sexes, early-adulthood CHX treatment abolished the normal age-related decline in spontaneous activity (Fig. 1e female, Supplementary Fig. 1c male) and learning and memory (Fig. 1f female, Supplementary Fig. 1d male). Cognitive function was assessed in flies with a well-established olfactory aversion training paradigm, where flies are trained to associate a particular odor with a negative sensation (electric shock)[18]. We confirmed that this result was not an artifact of early-life CHX treatment impacting sensorimotor responses to odors and electric shock at old ages (Supplementary Fig. 1f). We further evaluated gut health, specifically intestinal-barrier function, with the so-called "Smurf" assay[19]. Here, flies were fed non-absorbable blue dye, and loss of gut-barrier integrity was detected by counting the number of flies exhibiting blue dye outside the digestive tract and throughout the body (referred to as "Smurfs"). Early-life CHX treatment completely prevented age-dependent increases in Smurfs and drastically improved the gut-barrier integrity at old ages in both sexes (Fig. 1g female, Supplementary Fig. 1e male). Early-life CHX treatment also prevented reproductive senescence and significantly enhanced egg production at old ages (Fig. 1h); however, early-adulthood egg production was impaired and the fertility peak was delayed. These results imply a potential trade-off between early-adulthood fecundity and later health and longevity. As previously seen in lines selected for survival to reproduce at late ages[20], early-life reproduction is reduced while late-life reproduction is enhanced. Still, total lifetime egg production was not significantly affected (Supplementary Fig. 1h). Together, our data indicate that early-adulthood PT elevation, although beneficial for maximizing early-life fecundity, exerts long-lasting negative impacts on aging trajectories in later-life.

Ribosomal S6 kinase (S6K) promotes PT by phosphorylating/activating translational machinery[21]. He hypothesized that during the early-adulthood PT elevation, S6K phosphorylation will be increased. We thus overexpressed a dominant-negative/kinase-dead form of S6K ($S6K^{KQ}$)[22] during early-adulthood by using the GeneSwitch (RU486-inducible expression) system, which entirely blocked the early rise in PT (Fig. 2a, b). Results were remarkably similar to our CHX experiments. Early-adulthood $S6K^{KQ}$ overexpression increased lifespan to a similar extent as whole-adulthood $S6K^{KQ}$ overexpression (Fig. 2c). Further, early-adulthood $S6K^{KQ}$ overexpression also improved locomotor activity, cognitive function, gut-barrier integrity, and reproductive output at old ages in both sexes (Supplementary Fig. 2a–g, i, j). However, as with our CHX treatments, late-adulthood $S6K^{KQ}$ overexpression, did not significantly alter lifespan (Fig. 2c, Supplementary Fig. 2k). The inducing agent RU486 did not affect our experimental endpoints. Specifically, RU486 itself did not significantly alter lifespan at any adult stages in *daughterless-GeneSwitch* GAL4 (*daGS*)>$w^{1118}$ (control) flies (Supplementary Fig. 2l). Also, for *daGS* > $w^{1118}$, RU486 did not significantly affect egg production across ages (Supplementary Fig. 2h), indicating that RU486 itself did not delay the fertility peak or enhance reproductive fitness at old ages.

To further confirm that our findings are not from off-target effects of CHX, we used diazaborine (DAB) as our third method to prevent the early-adulthood PT elevation (Supplementary Fig. 1i, j). DAB is known to repress PT by blocking rRNA maturation and ribosomal biogenesis[8,23]. Consistent with our prior findings, early-adulthood DAB treatment significantly increased lifespan to a similar extent as whole-life treatment, with no longevity effects under late-life treatment (Supplementary Fig. 1k).

Life-/health-span benefits from blocking early-adulthood PT elevation were not due to dietary restriction. Neither early-life CHX treatment nor early-life $S6K^{KQ}$ overexpression had significant effects on

food intake and triglyceride/glycogen storage across ages (Supplementary Fig. 3d–i). Early-adulthood interventions also had no significant impacts on body size and size of internal organs such as muscle, gut, or fat body (Supplementary Fig. 3a–c). Unlike males, females treated with CHX/RU486 early in life did have lower body weight despite not eating less (Supplementary Fig. 3j, k), possibly due to impaired egg production during early-adulthood. However, since blocking the early rise in PT robustly prolonged lifespan in both sexes rather than just females, reduced body weight alone is unlikely a major factor contributing to the extended lifespan.

## Transiently inducing PT elevation in early-adulthood abolishes longevity benefits of insulin signaling mutants

Insulin/IGF signaling is a major growth-regulatory pathway, whose inhibition robustly extends lifespan and delays age-related diseases in multiple animal species[24]. We examined how PT dynamics are altered in insulin/IGF signaling-deficient dwarf flies (*chico* homozygotes). Mutations in *chico*, the *Drosophila* homolog of vertebrate insulin receptor substrate (IRS), have been shown to prolong life-/health-span and protect against neurodegeneration[25–27]. Long-lived *chico* homozygotes of both sexes, unlike wild-type counterparts, did not show the early-adulthood PT elevation (Fig. 2d, Supplementary Fig. 1l). Interestingly, *chico* heterozygotes, which are also long-lived but do not show dwarfism phenotypes seen in *chico* homozygotes[25], likewise did not show the early-life PT elevation (Fig. 2e, Supplementary Fig. 1m). Therefore, the loss of early-adulthood PT elevation in long-lived *chico* flies is independent of dwarf phenotypes.

To examine whether *chico* homozygotes achieve longevity by preventing the rise in PT during early-adulthood, we experimentally induced the early-life PT elevation by transiently overexpressing a constitutively-active forms of S6K ($S6K^{TE}$)[28] during early-adulthood (day 0–4) with RU486 (Fig. 2f). We overexpressed S6K with a phosphomimetic T398E mutation because S6K phosphorylation at T398 was increased during early-adulthood (Fig. 3a) and crucial for the early-life PT elevation. Immediately after RU486 removal, PT was decreased and restored to basal levels (Fig. 2g), suggesting that we were able to transiently manipulate PT to be elevated only during the narrow time window of early-adulthood. *Chico* homozygotes had a robustly improved lifespan; however, $S6K^{TE}$ overexpression for just 4 days significantly shortened their lifespan to that of controls and largely abolished their longevity benefits (Fig. 2h). These results suggest that loss of transient early-life PT elevation is critical for the longevity benefit of *chico* flies.

## Preventing age-related drop in PT causes accelerated aging

Together, our findings strongly suggest that the early-adulthood elevation in PT can cause age-related functional declines. We hypothesize that the repression of PT starting after day 2 is an adaptive process to reduce proteostatic burden and slow age-related functional decline. We found that T398 phosphorylation of S6K sharply declines after reaching a peak at day 2, maintaining low basal levels at day 15 (Fig. 3a). This implies that age-related decline in PT might be driven by loss of S6K phosphorylation at T398. Thus, to evaluate our hypothesis, we prevented the age-dependent decline in PT by overexpressing phosphomimetic $S6K^{TE}$ starting at day 2. In control flies, PT was decreased by ~70% within 15 days, but flies overexpressing $S6K^{TE}$ did not significantly reduce PT after the peak and maintained high levels of PT (Fig. 3b). Preventing the age-related drop in PT shortened lifespan by ~46% in both sexes (Fig. 3c, d). However, re-enabling flies to lower PT with age after early-adulthood and correcting PT dynamics via CHX (Fig. 3c, Supplementary Fig. 4a) completely restored lifespan to the control level (Fig. 3d). Inability to suppress PT after early-adulthood also significantly shortened healthspan in both sexes. Even at middle ages, flies overexpressing $S6K^{TE}$ after the PT peak exhibited deteriorations in spontaneous

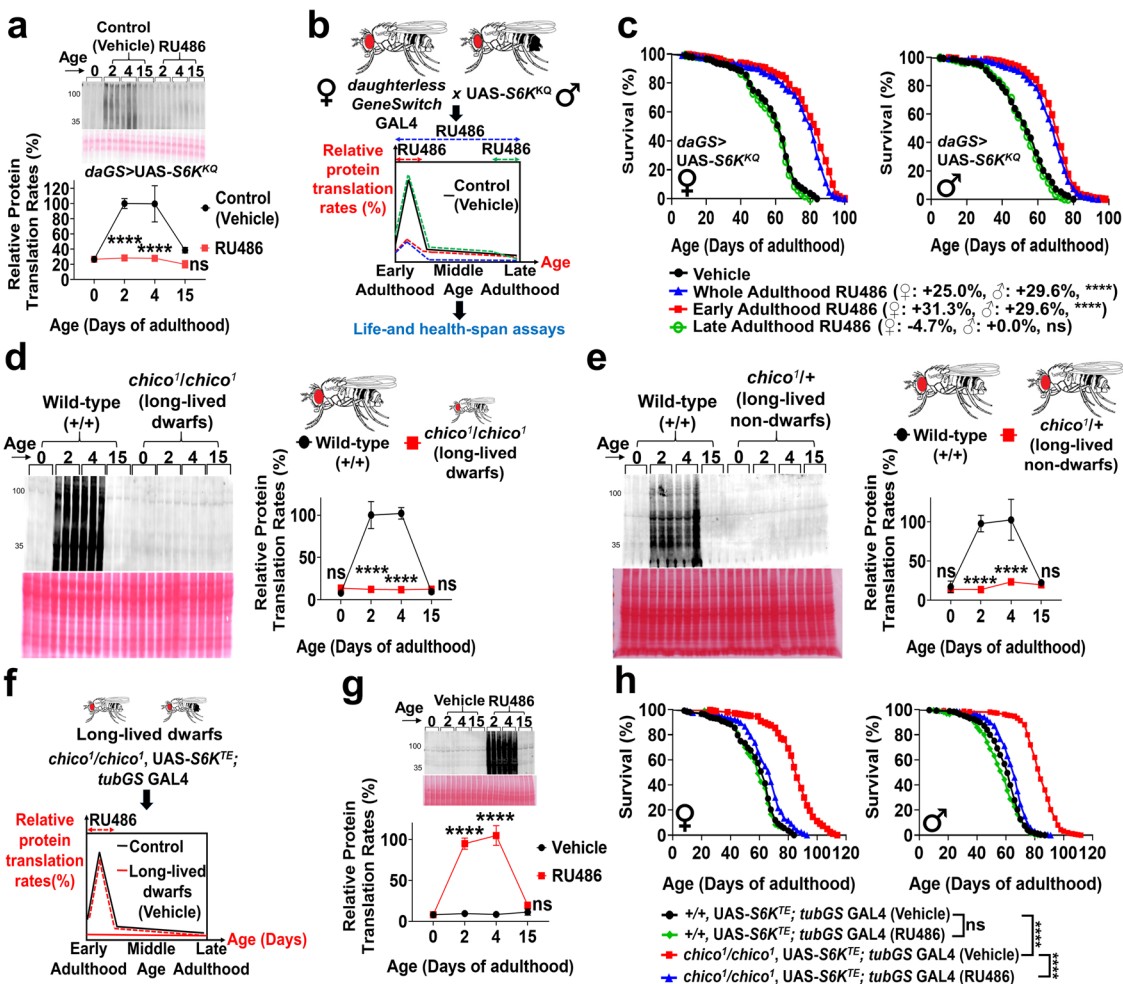

**Fig. 2 | Transiently inducing PT elevation in early-adulthood abolishes longevity benefits of dwarf flies. a** Early-adulthood PT in *daGS* > UAS-*S6K^KQ* flies after ± 200 μM RU486 (day 0–10), determined by puromycin incorporation normalized to Ponceau staining. *n* = 3/group; two-way ANOVA with Sidak post-hoc test. **b** Experimental scheme to transiently manipulate PT in different life stages; 200 μM RU486 given to *daughterless-GeneSwitch* GAL4 (*daGS*)>UAS-*S6K^KQ* flies during early-adulthood (day 0–10), late-adulthood (day 40–50), or whole-adulthood. **c** In *daGS* > UAS-*S6K^KQ* flies, early-adulthood (day 0–10) RU486 prolongs lifespan just like whole-adulthood RU486. Late-adulthood (day 40–50) RU486 does not alter lifespan. Each sex: *n* = 250/group; log-rank test. Puromycin incorporation in (**d**), *chico* homozygotes (female) and (**e**), *chico* heterozygotes (female) vs. wild-types. Absence of early-adulthood PT elevation in *chico* homozygotes and *chico* heterozygotes. *n* = 3/group; two-way ANOVA with Sidak post-hoc test. Male data in

Supplementary Fig. 1. **f** Experimental scheme to induce the early-adulthood PT elevation in *chico* homozygotes; ± 200 μM RU486 given to *chico^1/chico^1*, UAS-*S6K^TE*; *tubulin-GeneSwitch* (*tubGS*) GAL4 flies during early-adulthood (day 0–4). **g** Puromycin incorporation in *chico^1/chico^1*, UAS-*S6K^TE*; *tubGS* GAL4 flies (±RU486, day 0–4). *n* = 3/group; two-way ANOVA with Sidak post-hoc test. **h** Early-adulthood *S6K^TE* overexpression shortens lifespan and largely abolishes longevity of *chico* homozygotes. *chico* flies without early-adulthood *S6K^TE* inductions vs. controls. Female: + 34.4%, male: + 37.7% (% change in median lifespan); *chico* flies with early-adulthood *S6K^TE* inductions vs. controls. Female: + 6.3%, male: + 8.2%. Early-adulthood *S6K^TE* overexpression does not significantly alter lifespan of +/+ controls. Each sex: *n* = 250/group; log-rank test. Data shown as mean ± SD. **p* < 0.05, ***p* < 0.01, ****p* < 0.001, *****p* < 0.0001. Source data are provided as a Source Data file.

activity (Fig. 3e, Supplementary Fig. 4b), intestinal integrity (Fig. 3f, Supplementary Fig. 4f), learning and memory (Fig. 3g, Supplementary Fig. 4c–e), and signs of accelerated reproductive aging with total egg productions decreased by ~50% (Fig. 3h). In contrast, control flies barely showed functional impairments at this time point. Together, our data suggest that age-related decline in PT may not simply be a passive byproduct of aging but potentially an adaptive response to maximize reproductive output, slow functional decline, and promote healthy aging.

**Early-adulthood elevation in PT triggers protein aggregations at old ages**

Proteostasis deteriorates with age, causing accumulations of misfolded proteins and insoluble protein aggregates, which are heavily implicated in aging and age-related diseases[29]. We hypothesize that the early-adulthood PT surge and subsequent protein overload may

overwhelm the cellular folding capacity, facilitating protein misfolding and impairing proteostasis. To examine whether the early-adulthood PT elevation triggers age-dependent proteostatic dysfunction, we treated flies with CHX during early-adulthood and assessed the buildup of insoluble protein aggregates across ages. Remarkably, transiently blocking the early-adulthood elevation in PT completely prevented age-related accumulation of insoluble ubiquitinated proteins (IUP) (Fig. 4a). Conversely, long-lived *chico* homozygotes, which maintain a constant level of PT during their whole life, showed no signs of age-dependent IUP accumulation. However, transiently inducing the early-life PT elevation caused *chico* homozygotes to exhibit the age-related buildup of IUP just like wild-types (Fig. 4b). This is consistent with our prior finding that inducing the early-adulthood PT rise shortened the lifespan of *chico* homozygotes to that of controls. Together, our data suggest that the transient early-adulthood surge in PT triggers IUP accumulation at old ages.

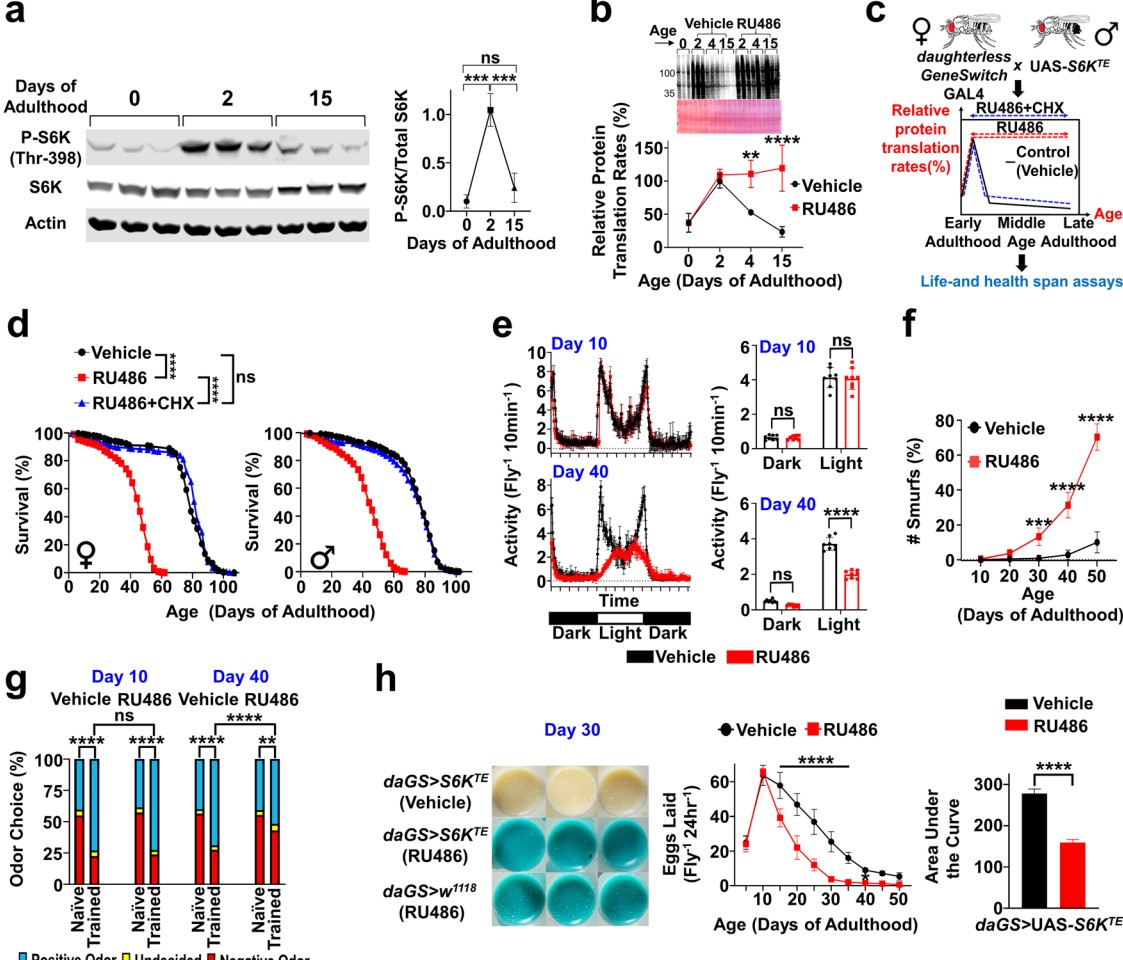

**Fig. 3 | Preventing age-related decline in PT causes accelerated aging and shortens lifespan and healthspan. a** Immunoblots of phospho-S6K and S6K. Actin was used as a loading control. $n = 3$; one-way ANOVA with Tukey's post-hoc test. **b** Puromycin incorporation in *daGS* > UAS-*S6K^{TE}* flies (±RU486 after day 2). $n = 3$/ group; two-way ANOVA with Sidak correction. **c** Experimental scheme to block age-related decline in PT; 200 μM RU486, 200 μM RU486 + 1 μM CHX, or vehicle given to *daughterless-GeneSwitch* GAL4 (*daGS*)>UAS-*S6K^{TE}* flies after day 2 (the PT peak). **d** *S6K^{TE}* overexpression after day 2 shortens lifespan (: −41.8%, male: −45.6%; % change in median lifespan), but concurrent CHX treatment restores lifespan comparable to that of controls. Each sex: $n = 250$/group; log-rank test. **e** *S6K^{TE}* overexpression after day 2 impairs locomotion at day 40. $n = 200$ female/group; two-way ANOVA with Sidak correction. **f** *S6K^{TE}* overexpression after day 2 causes

premature defects in gut-barrier integrity in Smurf assays. $n = 250$ female/group; two-way ANOVA with Sidak correction. **g** *S6K^{TE}* overexpression after day 2 impairs cognition in olfaction aversion training at day 40. $n = 200$ female/group; Chi-square test. **h** (Left) representative images of eggs laid on vials at day 30 by *daGS* > UAS-*S6K^{TE}* flies and *daGS* > *w^{1118}* (control) flies treated with ± RU486 after day 2. (Middle) *S6K^{TE}* overexpression after day 2 causes faster age-related decline in egg production. $n = 100$/group; two-way ANOVA with Sidak correction. (Right) Area under the curve was calculated to determine lifetime egg production. *S6K^{TE}* overexpression after day 2 impairs lifetime egg production. Two-tailed Student's t-test. Data shown as mean ± SD. *$p < 0.05$, **$p < 0.01$, ***$p < 0.001$, ****$p < 0.0001$. male healthspan data in Supplementary Fig. 4. Source data are provided as a Source Data file.

## Early-adulthood elevation in PT drives proteostatic dysfunction/aging via juvenile hormone

Notably, as shown in prior *C. elegans* studies, most proteins that comprise insoluble protein aggregates formed during aging are proteins known to function predominantly in *early*-adulthood[30]. In *Drosophila*, we thus hypothesize that aggregation-prone proteins greatly synthesized in early-adulthood under high PT may trigger age-dependent proteostatic dysfunction. To test this hypothesis and identify candidate aggregation-prone proteins, we used liquid chromatography/electrospray ionization-tandem mass spectrometry (LC/ESI-MS/MS). We examined how blocking the early-life elevation in PT impacts the proteome synthesized in early-adulthood and accordingly, components of insoluble protein aggregates at old ages.

During the early-life elevation in PT, large lipid transfer protein (LLTP) families such as vitellogenin (Vg)-1, 2 and apolipophorin (apoLp) were greatly synthesized along with proteins important for

lipid/carbohydrate metabolism, ATP generation, and reproduction/gametogenesis (Fig. 4c, Supplementary Data 1). LLTP families – lipid transfer proteins conserved across animal species including humans – play pivotal roles in reproduction and energy metabolism[31]. Meanwhile, proteins characteristic of larval fat bodies, such as larval serum proteins (LSP1-α, β, γ and LSP-2) and fat body proteins (FBP-1,2), were significantly downregulated (Fig. 4c, Supplementary Data 1). These results suggest that lipoprotein remodeling occurs during early-adulthood to facilitate *larval* fat body histolysis while developing the *adult* fat body where LLTPs are synthesized[32,33]. Early-adulthood CHX treatment completely reversed this lipoprotein remodeling, as manifested by reduced levels of LLTPs, sustained high LSP/FBP levels, and significantly affected lipid catabolism on Gene Ontology analysis (Fig. 4c, Supplementary Data 1, Supplementary Figs. 5, 6a, b). Early-adulthood CHX treatment caused widespread reductions in insoluble proteins at old ages, with LLTPs being the most significantly downregulated proteins (Fig. 4d, Supplementary Data 1). These results show

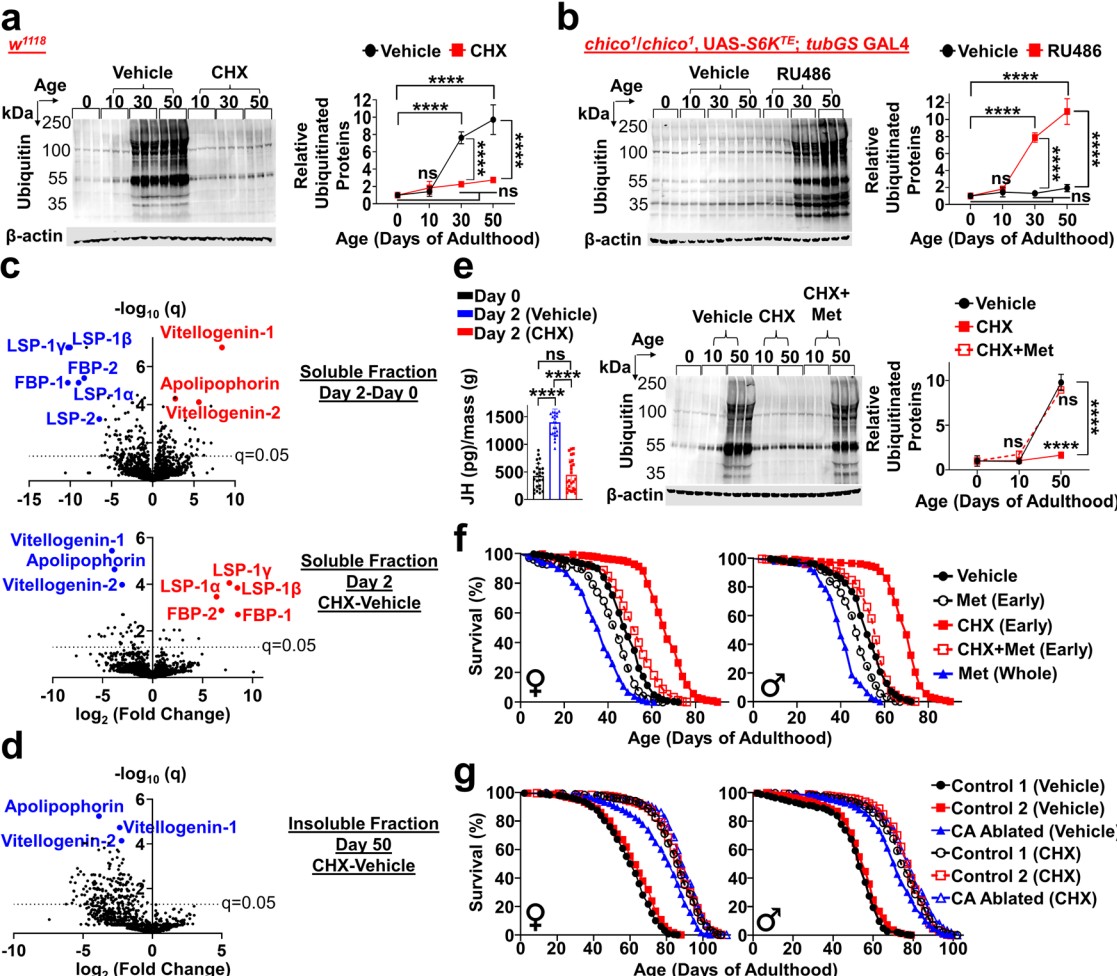

**Fig. 4 | Transient elevation in PT during early-adulthood triggers age-related proteostatic dysfunction and drives aging via juvenile hormone.** Early-adulthood=day 0–10. **a** Early-adulthood 1 μM CHX prevents age-related buildup of insoluble ubiquitinated proteins (IUP). $n = 3$/group; two-way ANOVA with Sidak correction. **b** No age-related buildup of IUP in long-lived *chico* homozygotes. $S6K^{TE}$ overexpression (day 0–4) causes age-related buildup of IUP in *chico* homozygotes. $n = 3$/group; two-way ANOVA with Sidak correction. **c** Volcano plots of proteomic analyses comparing Triton X-100 soluble fractions from day 2 vs. day 0 flies (top) and day 2 flies (±CHX, bottom). $n = 8$/group; two-tailed Student's t-test, FDR < 0.05. **d** Volcano plot of proteomic analysis comparing Triton X-100 insoluble fractions from day 50 flies (±CHX (day 0–10)). $n = 8$/group; two-tailed Student's t-test, FDR < 0.05. **e** (Left) Juvenile hormone levels measured via GC-MS normalized to wet mass in day 0 flies and day 2 flies treated with vehicle or CHX. $n = 25$/group. (Right) Early-adulthood CHX prevents buildup of IUP at day 50; however, early-adulthood

CHX+Met (methoprene, 25 μg/mL) causes buildup of IUP at day 50 just like controls. $n = 3$/group; two-way ANOVA with Sidak correction. **f** Early-adulthood Met (25 μg/mL) largely abolishes CHX-mediated longevity (female: + 6.1% vs. + 36.7%, male: + 7.7% vs. + 34.6%; % change in median lifespan, $p < 0.0001$). Early-adulthood Met alone shortens lifespan ($p < 0.0001$), but the effect is smaller than what we observe under simultaneous CHX treatment (proportional hazard analysis in Supplementary Fig. 6). Whole adulthood Met shortens lifespan ($p < 0.0001$). Each sex: $n = 250$/group; log-rank test. **g**, Lifespan extension by early-adulthood CHX is diminished under CA (corpora allata) ablation (female: + 7.9% vs. + 41.5%, male: + 9.7% vs. + 42.5%). Control 1=*Aug21*-GAL4 > $w^{1118}$; Control 2 = $w^{1118}$ > UAS-*NiPp1*; CA ablated=*Aug21*-GAL4 > UAS-*NiPp1*. Each sex: $n = 250$/group. Proportional hazard analysis in Supplementary Fig. 6. Data shown as mean ± SD. *$p < 0.05$, **$p < 0.01$, ***$p < 0.001$, ****$p < 0.0001$. Source data are provided as a Source Data file.

that LLTPs, prone to be aggregated at old ages, are increasingly synthesized when PT is elevated in early-adulthood, while early-adulthood CHX treatment blocks this.

Unlike in insoluble fractions, early-life CHX treatment did not cause dramatic proteome changes in the soluble fractions at old ages, except for upregulation of proteins that promote function of muscles, heart, and detoxification (Supplementary Fig. 7a). However, overexpression of these proteins did not significantly alter lifespan (Supplementary Fig. 7b–d), suggesting that these proteins are upregulated as a byproduct rather than a driver of healthy aging.

We note that juvenile hormone (JH), mainly synthesized in the corpora allata (CA), not only promotes larval fat body histolysis but also upregulates all LLTP families during early-adulthood[33]. With targeted MS/MS, we found that JH levels significantly increase from day 0 to day 2 when PT rises markedly (Fig. 4e). This rise in JH during

elevated early-adulthood PT was prevented by early-adulthood CHX treatment. For long-lived *chico* homozygotes with no apparent early-life PT elevation (Fig. 2), JH stays at low basal levels in early-adulthood[34]. We thus investigated if the early-life elevation in PT drives proteostatic dysfunction and age-related functional decline via JH. Surprisingly, after methoprene (Met; JH analog) treatments during early-adulthood, blocking the early-life PT elevation no longer produced any benefits in proteostasis at old ages (Fig. 4e). Early-adulthood Met also largely abolished longevity benefits conferred by blocking the early-life PT elevation (Fig. 4f; female: + 6.1% vs. + 36.7%, male: + 7.7% vs. + 34.6%). Early-adulthood Met alone shortened lifespan as well, but two-factor proportional hazard analysis showed the effect to be significantly smaller than what we observed under simultaneous CHX treatments (Supplementary Fig. 6c). Similarly, under conditions of JH-deficiency (CA ablation)[33], lifespan extension

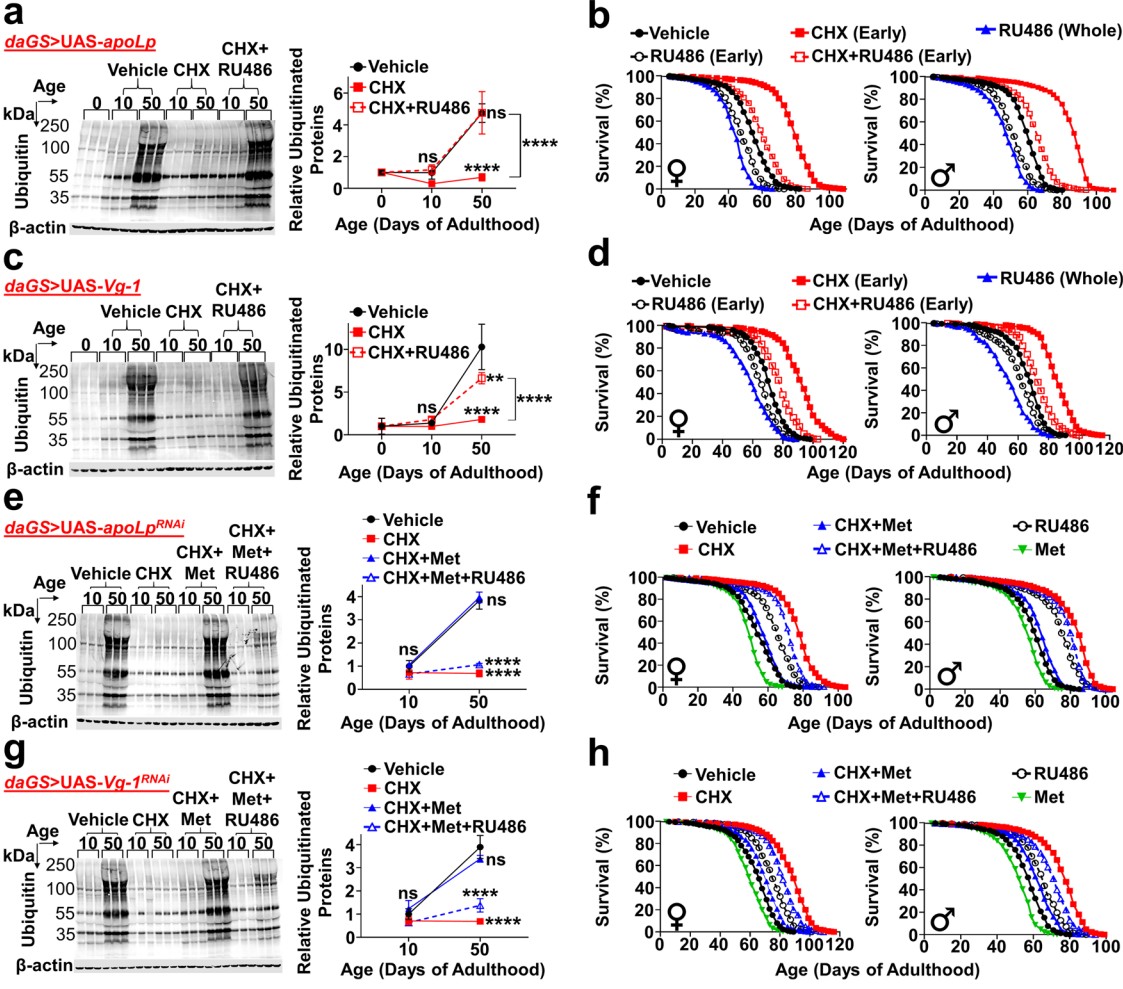

**Fig. 5 | Early-adulthood LLTPs are downstream targets of juvenile hormone in triggering proteostatic dysfunction and driving aging.** All treatments limited to early-adulthood (day 0–10) unless otherwise stated. **a** Early-adulthood *apoLp* overexpression abolishes CHX-mediated proteostatic improvements at day 50. **b** Early-adulthood *apoLp* overexpression largely abolishes CHX-mediated longevity (female: + 10.9% vs. + 45.5%, male: + 8.2% vs. + 44.3%, $p < 0.0001$). Early-adulthood *apoLp* overexpression alone shortens lifespan ($p < 0.0001$), but the effect is smaller than what we observe under simultaneous CHX (proportional hazard analysis in Supplementary Fig. 7). **c** Early-adulthood *Vg-1* overexpression largely abolishes CHX-mediated proteostatic improvements at day 50. **d** Early-adulthood *Vg-1* overexpression largely abolishes CHX-mediated longevity (female: + 9.6% vs. + 28.8%, male: + 7.4% vs. + 27.9%, $p < 0.0001$). Early-adulthood *Vg-1* overexpression alone shortens lifespan ($p < 0.0001$), but the effect is smaller than what we observe under simultaneous CHX (proportional hazard analysis in Supplementary Fig. 7).

**e** Early-adulthood Met (methoprene, 25 µg/mL) abolishes CHX-mediated proteostatic improvements at day 50; however, simultaneous *apoLp* knockdown during early-adulthood restores CHX-mediated proteostatic improvements at day 50. **f** Early-adulthood Met largely abolishes CHX-mediated longevity ($p < 0.0001$); however, simultaneous *apoLp* knockdown during early-adulthood largely restores the longevity ($p < 0.0001$). **g** Early-adulthood Met abolishes CHX-mediated proteostatic improvements at day 50; however, simultaneous *Vg-1* knockdown during early-adulthood restores CHX-mediated proteostatic improvements at day 50. **h** Early-adulthood Met largely abolishes CHX-mediated longevity ($p < 0.0001$); however, simultaneous *Vg-1* knockdown during early-adulthood largely restores the longevity ($p < 0.0001$). Immunoblots: $n = 3$/group; two-way ANOVA with Sidak correction. Lifespan experiments: $n = 250$/group (each sex); log-rank test. Data shown as mean ± SD. **$p < 0.01$, ****$p < 0.0001$. Source data are provided as a Source Data file.

by early-adulthood CHX treatment was significantly diminished (Fig. 4g. Supplementary Fig. 6d; female: + 7.9% vs. + 41.5%, male: + 9.7% vs. + 42.5%). Together, these data suggest that the early-life elevation in PT triggers age-dependent proteostatic dysfunction and drives aging primarily via JH.

Enhanced protein turnover by proteasomal degradation can improve proteostasis and lifespan[35]. Blocking the early-life elevation in PT slowed down age-related decline in proteasome activity by 58% (Supplementary Fig. 7e). This may be due to selective translational upregulation of proteasome subunits[36] or enhanced proteasome assemblies by preventing proteasome subunits/ATP-generating machinery from being trapped into insoluble fractions (Supplementary Fig. 6e). However, based on the profound impact of early-life JH on proteostasis, we decided to focus on JH signaling.

## LLTPs: downstream JH targets regulating proteostasis/lifespan

We next examined if LLTPs largely synthesized during the early-adulthood elevation in PT are drivers of age-related declines and proteostatic dysfunction at old ages. After *apoLp* overexpression during early-adulthood, CHX treatment no longer produced any benefits in proteostasis at old ages (Fig. 5a, Supplementary Fig. 7f). Likewise, longevity benefits conferred by early-life CHX treatment were largely abolished after *apoLp* overexpression during early-adulthood (Fig. 5b; female: + 10.9% vs. + 45.5%, male: + 8.2% vs. + 44.3%). Early-adulthood *apoLp* overexpression alone shortened lifespan as well, but proportional hazard analysis showed the effect to be significantly smaller than what we observed under simultaneous CHX treatments (Supplementary Fig. 7g). Similarly, early-adulthood *Vg* overexpression substantially abolished improvements in proteostasis/lifespan caused by blocking the early rise in PT (Fig. 5c, d, Supplementary Fig. 7f, g).

Further, knocking down *Vg* and *apoLp* during early-adulthood significantly prolonged lifespan in control flies but not in flies treated with CHX treatment in early life (Supplementary Fig. 7h–j). These data suggest that early-adulthood elevation in PT triggers age-dependent proteostatic dysfunction and drives aging primarily via aggregation-prone LLTPs.

To investigate if LLTPs are downstream targets of JH in mediating age-dependent proteostatic dysfunction/aging, we knocked down LLTPs while treating flies with CHX and Met during early-adulthood (Supplementary Fig. 7f). As previously shown, after early-adulthood Met treatments, CHX no longer produced any benefits in proteostasis at old ages; however, simultaneous *apoLp* knockdown during early-adulthood re-enabled flies to have improved proteostasis at old ages and prevented age-related accumulations of IUP (Fig. 5e). Similarly, although early-adulthood Met largely abolished longevity benefits conferred by blocking the early-life PT elevation, simultaneous *apoLp* knockdown during early-adulthood almost entirely restored lifespan extension effects (Fig. 5f). We observed similar results after *Vg* knockdown during early-adulthood (Fig. 5g, h). These results suggest that during the early-adulthood PT rise, JH impairs proteostasis at old ages and negatively impacts future lifespan mainly via LLTPs.

### Fat body remodeling alone is not sufficient for longevity benefits

Fat body maturation during early-adulthood requires JH and is responsible for LLTP production[32,33]. We thus hypothesized that eliminating the early-life PT surge may provide longevity benefits by mainly inhibiting fat body remodeling. To test this hypothesis, we repressed PT only in the fat body by overexpressing $S6K^{KQ}$ under the fat body-specific driver (*S106*-GS)[37]. This produced a ~90% reduction in PT in the fat body without impacting other tissues (Supplementary Fig. 8a). However, early-adulthood CHX treatment still produced a significant lifespan extension in both sexes (Supplementary Fig. 8b). We further tested this by ablating the fat body by overexpressing *reaper* (apoptosis inducer) under the fat body driver. Even after the fat body was ablated, CHX during early-adulthood still robustly improved lifespan to a similar extent (Supplementary Fig. 8c, d). We repeated these experiments with another fat body-specific driver (*S32*-GS)[37] and obtained similar results (Supplementary Fig. 8e, f). Early-adulthood CHX also did not significantly affect Oil Red O staining and triglyceride levels of fat body (Supplementary Fig. 8g). Likewise, inducing the early-adulthood PT elevation in *chico* homozygotes failed to stimulate fat body storage although it was sufficient to abolish their longevity benefits (Supplementary Fig. 8h). These results suggest that longevity benefits from blocking the early-adulthood PT elevation cannot be fully explained by fat body remodeling alone; tissues other than fat body are affected during early-life PT elevation to impact later-life aging trajectories. Of note, fat body is not the only major source for LLTPs; LLTPs are also predominantly synthesized in germline stem cells (GSCs), neurons, glia, cardiac muscles, nephrocytes, etc.[38].

We also demonstrated that early-adulthood CHX improves lifespan not by simply impairing reproductive functions of fat body proteins. Specifically, although overexpression of LLTPs during early-adulthood largely abolished CHX-mediated longevity, it did not restore impaired early-adulthood egg production (Supplementary Fig. 8i, j). This suggests that aggregation-prone properties of fat body proteins rather than their reproductive functions may be key to lifespan regulation.

### Early-adulthood elevation in PT silences stress responses via germline signaling

Since fat body remodeling alone was not sufficient to fully explain longevity benefits conferred by repressing the early-life PT elevation, we investigated alternative mechanisms. GSC proliferation peaks in early-adulthood to increase early-life reproductive fitness[39]. We hypothesized that the early-adulthood elevation in PT would impact

GSCs because it had profound effects on early-life egg production. Interestingly, in *C. elegans*, proliferative GSCs cause cellular stress responses (CSR) to sharply drop at the onset of reproductive maturity during early-adulthood by epigenetically silencing Hsp70 and Hsp16[40]. Indeed, inhibiting GSC proliferation or ablating GSCs prevents the precipitous decline in CSR during early-adulthood, enhances proteostasis at old ages, and improves lifespan[40–43]. We thus examined whether the early-adulthood elevation in PT triggers GSC-dependent silencing of CSR.

Consistent with prior *C. elegans* studies[40], heat/oxidative stress resistance sharply declined within 4 days of early-adulthood in *Drosophila*, which was entirely prevented by GSC ablation[42] (Supplementary Fig. 9a, b). Remarkably, as with GSC ablation, eliminating the early-life elevation in PT with CHX completely prevented the sharp drop in stress resistance during early-adulthood (Fig. 6a, b). To examine if the early-life elevation in PT transcriptionally remodels stress resistance, we performed RNAseq analyses on flies +/− oxidative stress (paraquat [PQ]) +/− early-adulthood CHX treatment. We observed that the transcriptional profile in response to PQ dramatically changed from day 0 to day 10; however, after early-adulthood CHX treatment, the transcriptional profile of day 10 flies became similar to that of day 0 flies (Fig. 6c). Day 0 flies, in response to PQ, upregulated CSR genes important for oxidative/heat stress resistance, detoxification, and xenobiotic metabolism while downregulating genes important for reproduction, iron metabolism, and extracellular matrix formation (Fig. 6c, Supplementary Fig. 10a, b). However, day 10 flies, in response to PQ, upregulated genes important for reproduction and extracellular matrix formation while downregulating CSR genes (Fig. 6c, Supplementary Fig. 11a, b). After blocking the early-life elevation in PT with CHX, day 10 flies, similar to day 0 flies, upregulated CSR genes while downregulating genes important for reproduction, iron metabolism, and extracellular matrix formation (Fig. 6c, Supplementary Fig. 12a, b). Early-adulthood CHX treatment also upregulated genes important for protein folding and microtubule remodeling while downregulating genes important for apoptosis, antibacterial responses, phospholipid metabolism, etc.

Consistent with our RNAseq data, under oxidative/heat stresses, induction of *sod-1*, *hsp-22*, and *hsp-23* genes became significantly attenuated during early-adulthood; however, both GSC ablation and blocking the early rise in PT entirely restored induction of these CSR genes (Fig. 6d, e, Supplementary Fig. 9c, d). Early-life CHX treatment appears to improve stress resistance by inhibiting GSC-dependent silencing of CSR genes. In support of this, blocking the early-life elevation in PT enhanced stress resistance and induction of CSR genes in GSC-intact flies but not in GSC-ablated flies (Supplementary Fig. 9a, b, Fig. 6d, e). Further, blocking the early-life elevation in PT significantly reduced GSC proliferation during early-adulthood (Supplementary Fig. 9e).

We then investigated whether the early rise in PT regulates aging via GSC signaling. In GSC-intact flies, blocking the early-life elevation in PT robustly improved lifespan (Fig. 6f; female: + 41.5%, male: + 42.5%). In GSC-ablated flies, early-life CHX treatment still provided a longevity benefit, but the size of this benefit is substantially less than that conferred in GSC-intact flies (Fig. 6f; female: + 7.9%, male: + 9.7%). Blocking the early rise in PT did not improve lifespan simply by inducing sterility and abolishing survival costs of producing gametes. We showed that early-adulthood CHX treatment prolonged lifespan in GSC-intact dominant female-sterile mutant lines ($Ovo^{D1}$)[44] to a similar, blocking the early-life elevation in PT significantly prolonged lifespan just like in fertile backgrounds (Fig. 6g). Our data suggest that the early-adulthood elevation in PT triggers transcriptional silencing of CSR genes and drives aging-related declines primarily via GSC signaling.

Taken together, our work proposes that the transient early-adulthood elevation in PT remodels later-life proteostasis network and

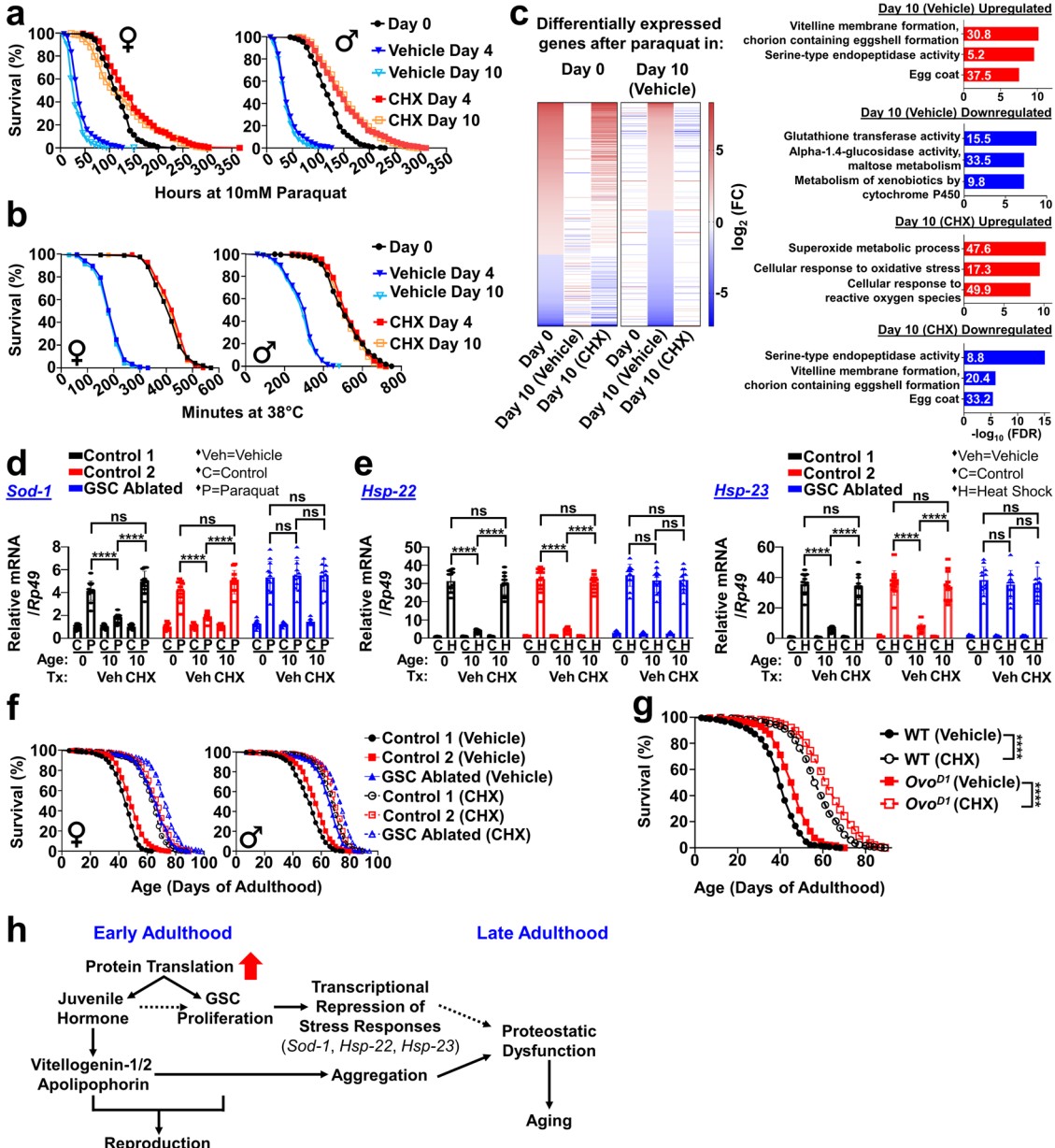

**Fig. 6 | Early-adulthood elevation in PT transcriptionally silences stress resistance genes and drives aging via germline stem cell signaling.** Early-adulthood CHX (1 μM) treatment prevents the sharp drop in (**a**), oxidative stress resistance ($p < 0.0001$) and **b**, heat stress resistance ($p < 0.0001$) during early-adulthood. Oxidative stress: 10 mM paraquat; heat stress: 38 °C. Each sex: $n = 250$/group; log-rank test. (**c**), (Left) Heatmap of differential gene expressions by RNAseq after 10 mM paraquat in day 0 flies and day 10 flies treated with vehicle or CHX. $n = 8$/group. FC=fold change. (Right) Top 3 GO biological processes enriched in genes that are up-/down-regulated after paraquat in day 10 flies treated with vehicle or CHX. White numbers in bar graphs denote fold enrichment. Early-adulthood CHX enhances (**d**), *Sod-1* inductions in response to 10 mM paraquat and (**e**), *Hsp-22/Hsp-* *23* inductions in response to 38 °C in germline stem cell (GSC)-intact flies but not in GSC-ablated flies. Control 1 = $w^{1118}$ > UAS-*bam*; Control 2 = *NGT*-GAL4 > $w^{1118}$; GSC ablated=*NGT*-GAL4 > UAS-*bam*. $n = 12$/group; two-way ANOVA with Sidak post-hoc test. **f** Lifespan extension by early-adulthood (day 0–10) CHX is diminished with GSC ablation (female: +10.6% vs. + 46.7%, male: + 12.1% vs. + 30.8%). Each sex: $n = 250$/group; log-rank test. **g** Early-adulthood CHX improves lifespan in both fertile controls and sterile *Ovo*[D1] mutants. WT Wild-types. $n = 250$/group; log-rank test. **h** Proposed model describing mechanisms by which elevated early-adulthood PT triggers proteostatic dysfunction and drives aging. Data shown as mean ± SD. *$p < 0.05$, **$p < 0.01$, ***$p < 0.001$, ****$p < 0.0001$. Source data are provided as a Source Data file.

alters future aging trajectories via two signaling pathways (Fig. 6h). We demonstrate that the early-adulthood elevation in PT enhances JH signaling, which upregulates LLTPs including vitellogenins and apolipophorins. These aggregation-prone LLTPs assist with reproduction but trigger proteostatic dysfunction at old ages. In parallel, the early-life elevation in PT enhances GSC proliferation, which transcriptionally silences CSR genes that can potentially be essential for proteostasis maintenance. We propose that these two pathways in tandem can drive age-related functional declines.

## Discussion

We demonstrate that the transient elevation in PT during early-adulthood exerts long-lasting *negative* impacts on aging trajectories by triggering proteostatic dysfunction at old ages. Our results suggest that the rapid suppression of PT after early-adulthood may be critical to alleviating late-life proteostatic burden, slowing down age-related functional decline, and improving life-/health-span. Our studies imply that the rise and fall in PT over time may impact aging in the opposite direction from what was previously assumed.

We propose the early-life elevation in PT to represent an antagonistic pleiotropic effect, producing beneficial effects in young adults but ultimately causing harmful effects at old ages and driving senescence. High PT in early life can promote reproduction by stimulating GSC proliferation and increasing levels of LLTPs via JH. GSC proliferation is essential for continuously producing gametes, and LLTPs are important for maturing sperms/eggs[45–47]. In late-adulthood, however, aggregation-prone LLTPs can trigger proteostatic dysfunction. Additionally, proliferating GSCs can increase susceptibility to proteostatic stress by silencing CSR genes, which may contribute to aging. Although we identified LLTPs and JH to be key molecules synthesized during the early-life PT elevation that can accelerate age-related declines, these factors alone were not responsible for all the longevity effects.

We demonstrated that LLTPs synthesized in early-adulthood trigger age-dependent proteostatic dysfunction; however, the exact underlying molecular mechanism still needs to be elucidated. LLTPs with highly-enriched β-sheet structures may act as seeds to initiate misfolding/aggregation of other native proteins or trap already-misfolded proteins[48]. Alternatively, LLTPs may allocate lipids (energy sources) into the germline at the cost of downregulating maintenance/repair in somatic tissues. Indeed, LLTPs are essential for shuttling lipids from somatic tissues, mainly intestines, to the germline and supporting reproduction[49]. Interestingly, after GSC ablation, intestinal lipid stores are no longer depleted with age, and mono-unsaturated fatty acids are generated from breakdown of excess fat, which act as bioactive lipid signals to improve proteostasis/lifespan[50–52]. Future lipidomic studies will help determine whether LLTPs synthesized in early-adulthood alter lipid partitioning/compositions to remodel proteostasis in somatic tissues.

Early-adulthood elevation in PT may be a driver of age-related decline in animal species other than *Drosophila*. For example, long-lived insulin/IGF-1 receptor *daf-2* mutant worms, similar to *chico* homozygote flies, maintain low PT during early-adulthood, unlike wild-types[2]. In addition, transient treatment of long-lived Ames dwarf mice with growth hormone (GH) in early postnatal life largely abolished their longevity[53,54]. Since GH is a major endocrine factor that stimulates PT and sharply rises during early-adulthood in mammals (including humans)[55], GH may regulate mammal longevity by modulating the early-adulthood elevation in PT. LLTPs synthesized during early-adulthood may also regulate age-related decline in higher-order animals. Centenarians showed low levels of LLTPs, which have been associated with reduced risk of age-related metabolic/cardiovascular diseases and once proposed as a longevity syndrome[56–58]. Further studies are needed to determine whether longevity regulation by the early-life PT elevation is conserved in other animal species.

Our work suggests that age-related decline in PT is unlikely a passive byproduct of aging but an *adaptive* response maximizing lifetime reproductive output and slowing age-related functional declines. PT begins to drop as early as day 2 post-eclosion, reaching low basal levels by day 15. Flies are still actively reproducing at these ages with peak egg production at ~day 10. When we blocked the age-related drop in PT, flies showed reduced lifespan and deteriorations in locomotion, cognitive functions, and gut-barrier integrity at earlier ages. They also experienced accelerated reproductive aging with reductions in total egg productions by ~50%. Since age-related drop in PT occurs before the fertility peak and can have positive impacts on reproductive outcome, we hypothesize that animals are under active selection to repress protein translation to delay functional declines and extend their reproductive period.

Our findings support a view that long-term health and aging trajectories can be modulated by transient biological changes occurring early in life. This study provides a theoretical framework for understanding how the rise and fall in PT determines aging trajectories. Our work also provides a foundation for future research, including whether high early-adulthood PT is a potential early biological event/marker driving age-related diseases.

## Methods

### *Drosophila* stocks, husbandry, and lifespan assays

*chico*[1]/*chico*[1] was provided by Dr. Marc Tatar; $w^{1118}$ and *Daughterless-GeneSwitch*-GAL4 (*daGS*) were provided by Dr. Scott Pletcher; *Tubulin-GeneSwitch*-GAL4 (*tubGS*) was provided by Dr. David Walker; UAS-*apoLp* was provided by Dr. Joaquim Culi; UAS-*bam* was provided by Dr. Erika Bach; UAS-*Stretchin-Mlck* was provided by Dr. Mark VanBerkum. The rest of fly lines were obtained from the Bloomington Drosophila Stock Center and FlyORF. *ovo*$^{D1}$ and UAS/GAL4 lines were backcrossed to $w^{1118}$ for 8–10 times. Transgene expression was confirmed by qRT-PCR. All fly lines were maintained in a humidified 25 °C incubator with 12 h light/dark cycles. Flies were developed on agar-cornmeal-dextrose-yeast growth (CT) media[59] and transferred to 10% sugar/yeast (SY10) media after eclosions[60]. Flies were allowed to mate for 48 h, separated/sorted into females and males, and maintained as separate sexes in SY10 vials (25 flies/vial) from then onward. For life-span assays, 10 vials/group for each sex were used. Every 2–3 days, flies were transferred to fresh SY10 media without gassing and survival was scored. dLife software[61] was used to record survival and analyze median/maximum lifespan via logrank analysis. Vials were randomized in terms of tray position and semi-blinded to reduce impacts of environment/investigator bias.

### Stress resistance assays

Before stress challenges, flies were pre-treated with 1 μM cyclohex-amide (CHX; ChemService)/water (vehicle) dissolved in SY10 for 4 or 10 days. For oxidative stress resistance assays[62], flies were transferred onto vials containing half a Kimwipe soaked in 1 mL of 5% sucrose and 10 mM paraquat (Sigma). Flies were transferred to fresh vials every 8 h containing paraquat/sucrose medium, dead flies were scored after each transfer. For heat stress resistance assays[63], flies were exposed to heat (38 °C) by sinking vials in a water bath. Mortality was recorded at 30 min intervals. For both stress resistance assays, 10 vials (25 flies each)/group were used for each sex. For qRT-PCR of stress resistance genes and RNAseq, flies were treated with paraquat for 12 h or exposed to 38 °C for 1 h ($n = 12$/group for each sex).

### Olfactory aversion training and sensorimotor response assays

As previously[18], flies were exposed (via an air pump) in alternation to 2 neutral odors (3-octanol [OCT] and 4-methylcyclohexanol [MCT], 1/10 dilution in mineral oil). Training involved 3 training rounds with alternation odors, 5 min per round and exposure to a 100 V 60 Hz shock with one of the odors. All treatments performed under low red-light. The odor associated with the electric shock was alternated between vials. After training, flies were given 1 hr to recover and then placed in a T-maze (Celexplorer labs). Opposing odors were pumped from opposite sides of the maze. Flies were allowed 2 min to explore the maze, after which the maze sections were sealed and the number of flies in each chamber was scored. Before the training, natural odor preference was determined by exposing flies to 2 odors for 2 min. Flies were then trapped in their respective T-maze arms, anesthetized, and counted. Results were evaluated by Chi-square analysis. For each group, $N = 50$ flies x 4 vials/age were used.

As control measures, sensorimotor responses to the odors and electric shock were determined in a separate cohort[64]. Odor-avoidance responses were quantified by exposing naïve flies to one of the two odors (OCT or MCT) vs. air in the T-maze. The ability to sense and escape from electric shock (shock reactivity) was quantified in naïve flies by inserting electrifiable grids into both arms of the T-maze but delivering shock pulses in only one arm of the T-maze and allowing flies to choose between the two arms. After 2 min the chamber doors

were closed and the number of flies in each arm counted. For each group, $N = 50$ flies x 4 vials were used.

## Smurf assays

As previously done[19], flies were aged on regular SY10 medium until the day of the Smurf assay. Dyed SY10 medium was prepared by adding Blue Dye #1 (FD&C) at a concentration of 2.5% (wt/vol). Flies were maintained on dyed medium for 9 h. Flies were scored as a Smurf when dye coloration could be observed outside the digestive tract. For each sex, $N = 25$ flies x 10 vials/group were used.

## Spontaneous activity assay

Spontaneous activity was evaluated using a Drosophila activity monitor (Trikenetic). Activity was recorded in a humidified 25 °C incubator with 12 h light/dark cycles. Flies were allowed to acclimate for 8 h prior to data collection. Total activity quantified per 12 h cycle and divided by fly number. For each sex, $n = 10$ vials (20 flies per vial).

## Feeding assay and body weight measurements

Feeding was measured based on the consumption/excretion of dye-containing food[65]. SY10 media containing 1% w/v Blue Dye #1 (FD&C) was dispensed into plastic caps that fit wide plastic vials. After incubation at 25 °C for 24 h, food caps and flies were removed. Flies were homogenized in 1.5 mL of water to collect consumed dye. Samples were then centrifuged to remove pellet debris. The dye excreted by flies on the walls of the vials (excreted vial dye, ExVial) was collected by adding 3 mL of water to vials, followed by vortexing. Absorbance of the INT and ExVial dyes in water extracts was determined at 630 nm in a plate reader (SpectraMax iD3). Absorbance was converted to volumes of media consumed by interpolation from a standard curve. Extracts of flies fed media without dyes were used to control for background absorbance. For each sex, $N = 15$ flies x 10 vials/group were used. For measuring body weight, 10 flies were treated with ±1 μM CHX (ChemService)/± 200 μM RU486 (Cayman Chemical) for 10 days, collected with pre-weighted 1.5 mL Eppendorf tubes, and weighed with a precision microbalance. For each sex, 10 tubes/group were used.

## Fecundity assays

10 females and 10 males were mated for 2 days after eclosions. Females were then separated and maintained as separate sexes in SY10 vials ($n = 20$ vials/group; 5 females/vial). Fecundity was assayed as the number of eggs laid per female in each vial after 24 h period at the indicated age across the lifespan, counted under a stereomicroscope.

## BrdU labeling and immunostaining of ovarioles

BrdU staining was done as described previously in ref. 66. Ovaries of flies treated with 1 μM CHX/vehicle (day 0–5) were dissected and teased apart in an alternating order in Grace's medium (Lonza) and incubated in the same media containing BrdU (1 mg/mL, Sigma) at 25 °C for 1 h. After 3 washes with PBT (10 mM $NaH_2PO_4$/$Na_2HPO_4$, 175 mM NaCl, pH 7.4, 0.1% Triton X-100), ovaries were fixed with 4% paraformaldehyde for 10–15 min at room temperature. After blocking with 5% normal goat serum in PBT for an hour, primary and secondary antibodies were added. Antibodies used were: mice anti-BrdU (G3G4, DSHB, 1:50), rat anti-Vasa (a gift from Dr. Paul Lasko; 1:50), Alexa Fluor 488-conjugated goat anti-mouse and Alexa Fluor 568-conjugated goat anti-rat secondary antibodies (Thermo Scientific, 1:300). Samples were mounted in Vectashield with DAPI (Vector Laboratory) and imaged by Carl Zeiss LSM 700 confocal microscope. For co-localization analysis, JACoP plugin of ImageJ was used. After adjusting the threshold in different channels, the % area of co-localization was obtained.

## Triglyceride and glycogen measurements

As previously done[67], flies were homogenized in 500 μL of PBST (0.05% (v/v) Tween 20 in PBS) using a plastic pestle motor mixer. 200 μL of lysate was incubated at 70 °C for 5 min, chilled on ice, and incubated with 1 μL of 25 KU/mL lipoprotein lipase from Chromobacterium viscosum (Calbiochem) at 37 °C overnight. Debris was removed by centrifugation at 14,000 rpm for 3 min. Triglyceride content was determined by mixing 15 μL of the supernatant with 150 μL Free Glycerol Reagent (Sigma), incubating at 37 °C for 6 min, and measuring the absorbance at 540 nm. Triglyceride content was normalized to total protein levels measured by Bradford assay at 595 nm. To measure triglyceride levels in the fat body, flies were dissected in cold PBST, and collected fat bodies were processed with same procedures, as described above. For determining glycogen levels, 30 μL of the supernatant (after lipoprotein lipase treatment) was treated with 14 Units of amyloglucosidase (Sigma) at 50 °C for 1 h. 15 μL of the treated mix was combined with 150 μL of glucose reagent (Sigma), incubated at 37 °C for 30 min, and absorbance was measured at 340 nm. Glycogen content was also normalized to total protein. Triplicates of 10 flies/age group were used.

## Oil Red O staining

As previously done[68,69], intact fat body/carcasses were dissected in PBS and fixed in 4% paraformaldehyde for 20 min, then washed 3 times with PBS, incubated for 20 min in fresh Oil Red O solution (6 mL of 0.1% Oil Red O in isopropanol and 4 mL distilled water and passed through a 0.45 μM syringe), followed by rinsing with distilled water. Samples were mounted in Vectashield with DAPI (Vector Laboratory) and imaged by light microscopy. Staining was quantified by Image J.

## $^{35}$S-methionine incorporation assays

As previously done[70], flies aged on regular SY10 medium were placed and incubated on the SY10 medium supplemented with 2.5 μCi $^{35}$S-methionine (American Radiolabeled Chemicals)/mL of food for 24 h. 15 flies were frozen and then ground in 500 μL 1% SDS, boiled for 5 min. Samples were then centrifuged for 5 min at 10,000 g and supernatant was isolated. The protein was precipitated with 5% TCA and kept on ice for 1 h. Samples were centrifuged at maximum speed for 5 min, and the pellet was washed twice with ice cold 95% ethanol. The pellet was resuspended in 100 μL 1% SDS, and the protein content was measured using BCA Protein Assay (Pierce, Rockford, IL). The $^{35}$S radioactivity was measured using liquid scintillation (Beckman, Fullerton, CA). $^{35}$S incorporation was calculated by normalizing $^{35}$S counts/min to total protein levels.

## Puromycin incorporation assays

As previously done[17], flies aged on regular SY10 medium were placed and incubated on the SY10 medium supplemented with 600 μM puromycin (Fisher Bioreagents) for 24 h. 15 flies were frozen and ground in 160 μL RIPA buffer supplemented with protease and phosphatase inhibitors at 4 °C. The homogenate was centrifuged at maximum speed for 15 min and supernatant was isolated. Lysate protein content was determined using BCA Protein Assay (Pierce, Rockford, IL). Samples were mixed with Laemmli sample buffer supplemented with 5% β-mercaptoethanol and boiled at 95 °C for 5 min prior to immunoblotting. Normalized samples were run on SDS-10% PAGE gels, followed by standard Western blotting procedures. Total protein loading was visualized via Ponceau S (Sigma) staining. Anti-mouse puromycin antibody [3RH11] (Kerafast, 1:1000) and goat anti-mouse IgG polyclonal antibody (IRDye 800CW, LI-COR, 1:5000) were used. Imaging and quantification were performed using the Odyssey system (LI-COR) and Image J. Protein translation rates were determined by normalizing puromycin incorporation to Ponceau staining.

## In vitro translation assays

As previously done[17], whole bodies of flies were homogenized by a plastic pestle in 10 mM HEPES (pH 7.4), 5 mM DTT, and 1x complete protease inhibitor cocktail (Roche) and centrifuged at 14,000 g at 4 °C

for 15 min. 0.15 U/µl Micrococcal nuclease (New England BioLabs) and 1 mM CaCl$_2$ were added to the supernatant and incubated at 20 °C for 4 min, followed by the addition of 2 mM EGTA. Translation reaction mixtures contained 40% (v/v) extract, 2 nM luciferase RNA, 100 µM amino acid mix (Promega), 50 mM potassium acetate, 2.5 mM magnesium acetate, 0.1 mM spermidine, 20 U of RNase inhibitor (Fisher), 20 mM creatine phosphate (Roche Applied Science), and 80 ng/µl creatine kinase (Roche Applied Science). The mixture was incubated at 26 °C in a SpectraMax iD3 plate with luminescence measured every 5 min. The slope of the linear part of the luciferase activity curve was used to calculate the rate of in vitro protein translation.

### Plate-based proteasome activity assays
Whole bodies of 2 flies ($n = 12$/group) were homogenized by a plastic pestle in 100 µL chilled proteasome buffer (50 mM Tris, 5 mM MgCl2, 1 mM DTT, pH 7.4). Samples were then vortexed and centrifuged at 21,000 g at 4 °C for 15 min. In a black 96-well plate, 10 µL of supernatant was added to 80 µL proteasome buffer supplemented with additional 5 mM ATP for measuring 26 S proteasome activity. Finally, 50 µM Suc-LLVY-AMC fluorogenic substrate in 10 µL proteasome buffer was added to each well to measure chymotrypsin-like activity. The plate was incubated at 37 °C in a SpectraMax iD3 plate reader for 4 h with fluorescence measured every 10 min with 355 nm excitation and reading at 460 nm emission. Total proteasome activity per sample was defined as the delta between initial and final measurements.

### RNA extraction, cDNA synthesis, and qRT-PCR
RNA was isolated from flies using a standard Trizol method with 2 whole animals or 20 heads ($n = 12$/group). RNA quantified by Nano-Drop. cDNA was synthesized using the High-Capacity cDNA reverse transcriptase kit (Applied Biosystems) in technical triplicates, following the manufacturer's instructions. SYBR Green PCR Master Mix (Applied Biosystems) was used, following the manufacturer's instructions. Expressions of target genes were measured and normalized to *Rp49*. Primers used were *Rp49*: 5′-CGCTTCAAGGGACAGTATCTG-3′ and 5′-AAACGCGGTTCTGCATGA-3′; *Sod-1*: 5′-GGACCGCACTTCAATCCGTA-3′ and 5′-TGGAGTCGGTGATGTTGACC-3′; *GstD-1*: 5′-CGCGCCATCCAGGTGTATTT-3′ and 5′-CTGGTACAGCGTTCCCATGT-3′; *GstE-1*: 5′-ATAATGGCACTTTCATCTGGGAC-3′ and 5′-CACTGGCATCGAAGAAGAGAC-3′; *Catalase*: 5′-GATGCGGCTTCCAATCAGTTG-3′ and 5′-GCAGCAGGATAGGTCCTCG-3′; *TrxR-1*: 5′-TGGAGTCGGTGATGTTGACC-3′ and 5′-GAAGGTCTGGGCGGTGATTG-3′; *Hsp-22*: 5′-GCCTCTCCTCGCCCTTTCAC-3′ and 5′-TCCTCGGTAGCGCCACACTC-3′; *Hsp-23*: 5′-GGTGCCCTTCTATGAGCCCTACTAC-3′ and 5′-CCATCCTTTCCGATTTTCGACAC-3′; *Hsp-68*: 5′-GAAGGCACTCAAGGACGCTAAAATG-3′ and 5′-CTGAACCTTGGGAATACGAGTG-3′; *Hsp-70*: 5′-AGCCGTGCCAGGTTTG-3′ and 5′-CGTTCGCCCTCATACA-3′; *Stretchin-Mlck*: 5′-TCACTGGCGGAGAACTGTTC-3′ and 5′-CGGGTGCTTTCGTATCCAGT-3′; *Pdh*: 5′-ATAGCCAAGACCTTCGGTAACA-3′ and 5′-GCACTCAGTGTGGAATTGATGAT-3′; *apoLp*: 5′-TTGGAATCCTAGCTTCTGTGCT-3′ and 5′-AGTCATAGTAGTTGCCGGGTAT-3′; *Vit-1*: 5′-CAACTCCGTCAACCAGGCATT-3′ and 5′-GACAGGTGGTAGACTTGCTGC-3′; *Vit-2*: 5′-GCACCCTTTGCGTTATGGC-3′ and 5′-TAGAGCTTGTCCAACAGCGTA-3′

### Sequential protein extractions and protein aggregation assays
As previously done[17], 15 flies/age group were freshly prepared and homogenized in 200 µL of buffer A (20 mM Tris-HCl [pH 8.0], 150 mM NaCl, 1 mM EDTA, 1% Triton X-100, 1X Protease and Phosphatase Inhibitor Cocktails [Fisher Scientific]). Lysates were vortexed and centrifuged at 21,000 g for 15 min at 4 °C. Supernatants represented the detergent-soluble protein fraction. Pellets were washed in 500 µL buffer A and vortexed and centrifuged at 21,000 g for 15 min at 4 °C. Pellets were resuspended in 100 µL of buffer B (10 mM Tris-HCl [pH 7.5], 2% SDS, 1X Protease and Phosphatase Inhibitor Cocktails), sonicated for 1 min (6 x 10 s pulses), and incubated at room temperature

for 15 min. Extracts were cleared by centrifugation at 500 g for 10 min, yielding detergent-insoluble fractions. The total protein concentration in extracted fractions was determined by using BCA Protein Assay (Pierce, Rockford, IL), and normalized samples were run on SDS-10% PAGE gels, followed by standard Western blotting procedure. Antibodies used were: Rabbit anti-ubiquitin antibody (Cell Signaling, 1:1000), rabbit anti-β-actin antibody (Cell Signaling, 1:1000), and goat anti-rabbit IgG polyclonal antibody (IRDye 680RD, LI-COR, 1:5000). Imaging and quantification were performed using the Odyssey system (LI-COR) and ImageJ, and aggregate levels were calculated as ubiquitin signals normalized by actin levels.

### Juvenile hormone measurements
Total juvenile hormone content was quantified for whole bodies of day 0 flies and day 2 flies treated with ± 1 µM CHX for 2 days post-eclosion. For each group, 25 samples were collected, each consisting of 75 flies homogenized in 1.5 ml 90% HPLC grade methanol. Samples stored at −80 °C until analysis. JH was extracted with increasingly polar solutions of hexane, column purified, and converted to a d3-methoxyhydrin derivative. Total JH was measured using an established GC-MS method[71,72], conducted on an Agilent 8890 Series GC, equipped with a 30 m x 0.25 mm ZB-Wax GC column (Phenomenex), and coupled Agilent 7000D triple quadrupole mass spectrometer. Sample peaks were analyzed monitoring for m/z 76 and 225 to insure specificity to JHIII. Total abundance of JHIII was calculated against a standard curve and normalized to wet masses.

### Proteomic analysis
**Sample preparation.** Detergent-soluble-/-insoluble protein extracts were isolated, as described above. For each age, $n = 8$/group (each derived from whole bodies of 15 female flies) was used. Proteomics analysis was carried out as previously referenced[73] with minor changes, under section 2.5 nLC-ESI-MS2 under Protein IDs for GeLC. All proteins extracts were quantified using Pierce BCA Protein Assay Kit (Thermo Fisher Scientific), and a set amount of protein per sample was diluted to 35 µL using NuPAGE LDS sample buffer (Invitrogen). Proteins were then reduced with DTT and denatured at 70 °C for 10 min prior to loading everything onto Novex NuPAGE 10% Bis-Tris Protein gels (Invitrogen) and separated (35 min at constant 200 V). The gels stained overnight with Novex Colloidal Blue (Invitrogen). Following de-staining, each lane was cut into multiple MW fractions (3–8 fractions) and equilibrated in 100 mM ammonium bicarbonate (AmBc). Each gel plug was then digested overnight with Trypsin Gold, Mass Spectrometry Grade (Promega), following manufacturer's instruction. Peptide extracts were reconstituted in 0.1% Formic Acid (FA)/ ddH2O at 0.1 µg/µL.

**Mass spectrometry.** Peptide digests (8 µL each) were injected onto a 1260 Infinity nHPLC stack (Agilent Technologies) and separated using a 75 micron I.D. x 15 cm pulled tip C-18 column (Jupiter C-18 300 Å, 5 micron, Phenomenex). This system runs in-line with a Thermo Orbitrap Velos Pro hybrid mass spectrometer, equipped with a Nanospray FlexTM ion source (Thermo Fisher Scientific), and all data were collected in CID mode. The nHPLC is configured with binary mobile phases that includes solvent A (0.1% FA in ddH2O), and solvent B (0.1% FA in 15% ddH2O/85% ACN), programmed as follows; 10 min @ 5% B (2 µL/ min, load), 90 min @ 5–40% B (linear: 0.5nL/min, analyze), 5 min @ 70% B (2 µL/ min, wash), 10 min @ 0% B (2 µL/min, equilibrate). Following parent ion scan (300–1200 m/z @ 60 k resolution), fragmentation data (MS2) was collected on the most intense 15 ions. For data dependent scans, charge state screening and dynamic exclusion were enabled with a repeat count of 2, repeat duration of 30 s, and exclusion duration of 90 s.

**MS data conversion/searches.** XCalibur RAW files were collected in profile mode, centroided, and converted to MzXML using ReAdW

v. 3.5.1. mgf files were then created using MzXML2Search (included in TPP v. 3.5) for all scans. The data were then searched using SEQUEST (Thermo Fisher Scientific), which is set for three maximum missed cleavages, a precursor mass window of 20ppm, trypsin digestion, variable modification C @ 57.0293, and M @ 15.9949 as a base setting, with additional PTM's (ex: Phos, Ox, GlcNAc, etc.) that may be applied at a later time as determined to be of importance experimentally. Searches were performed with a species specific subset of the UniProtKB database.

**Peptide filtering, grouping, and quantification.** The list of peptide IDs generated based on SEQUEST search results was filtered using Scaffold (Protein Sciences, Portland Oregon). Scaffold filters and groups all peptides to generate and retain only high confidence IDs and generated normalized spectral counts (N-SC's) across all samples for the purpose of relative quantification. The filter cut-off values were set with minimum peptide length of > 5 AA's, with no MH + 1 charge states, with peptide probabilities of > 80% C.I., and with the number of peptides per protein ≥ 2. The protein probabilities were set to a > 99.0% C.I., and an FDR < 1.0. Scaffold incorporates the two most common methods for statistical validation of large proteome datasets, the false discovery rate (FDR) and protein probability[74–76]. Relative quantification across experiments was then performed via spectral counting[77,78], and when relevant, spectral count abundances were then normalized between samples[79].

**Systems/statistical analyses.** Gene ontology assignments and pathway analysis were carried out using MetaCore (GeneGO Inc., St. Joseph, MI). Interactions identified within MetaCore are manually correlated using full text articles. Detailed algorithms have been described previously in refs. 80,81. For the proteomic data generated, two separate non-parametric statistical analyses were performed between each pairwise comparison. These non-parametric analyses include 1) the calculation of weight values by significance analysis of microarray (SAM; cut off >|0.8| combined with, 2) T-Test (two tail, unequal variance, cut off of $p < 0.05$), which are then sorted according to the highest statistical relevance in each comparison. For SAM[82,83], whereby the weight value (W) is a statistically derived function that approaches significance as the distance between the means ($\mu 1$-$\mu 2$) for each group increases, and the SD ($\delta 1$-$\delta 2$) decreases using the formula, $W = (\mu 1$-$\mu 2)/(\delta 1$-$\delta 2)$. For protein abundance ratios determined with N-SC's, we set a 1.5–2.0 fold change as the threshold for significance, determined empirically by analyzing the inner-quartile data from the control experiments using ln-ln plots, where the Pierson's correlation coefficient (R) is 0.98, and > 99% of the normalized intensities fell between the set fold change. In each case, all three tests (SAM, T-test, or fold change) have to pass in order to be considered significant.

**Statistical analyses**
The Prism software package (Graphpad Software 8) and the Microsoft Office 2016 Excel software package were used to carry out statistical analyses. For all statistical analyses, a 2-sided $p < 0.05$ was accepted as statistically significant. All analyses were adjusted for multiple comparisons. Detailed information about statistical tests, $p$-values, and $n/N$ numbers are provided in the respective figures, figure legends, and methods.

**Reporting summary**
Further information on research design is available in the Nature Portfolio Reporting Summary linked to this article.

## Data availability
The Proteomic data generated in this study have been deposited in the PRIDE database under accession code PXD044118, https://doi.org/10.6019/PXD044118. The RNAseq data generated in this study have been deposited in the Gene Expression Omnibus database under accession code GSE239506. All other data supporting the findings of this study are available within the article and Supplementary Information files and from the corresponding author upon request. Source data are provided with this paper.

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

## Acknowledgements

We thank Marc Tatar, Scott Pletcher, David Walker, Erika Bach, Joaquim Culi, Mark VanBerkum, Bloomington Drosophila Stock Center, and FlyORF for fly stocks. We thank Paul Lasko for providing us antibodies. This research was supported by NIA/NIH R56-AG061051 & RF1 AG065301 (A.P.), Glenn Foundation for Medical Research, (A.P.), American Federation for Aging Research (AFAR) Grants for Junior Faculty (A.P.), Voelcker Young Investigator Award (A.P.), San Antonio Nathan Shock Center (A.P.), San Antonio Pepper Center (A.P.), Barshop Institute T32 Program on the Biology of Aging NIA T32-AG021890 (H.K., E.M.), UAB Medical Scientist Training Program NIH T32-GM008361 (H.K.), and South Texas Medical Scientist Training Program NIH T32-GM113896 (H.K.). The fly images were used under the Creative Commons license: "Characteristics and traits: Figure 10" by OpenStax College, Biology, CC BY 4.0: https://openstax.org/books/biology/pages/12-2-characteristics-and-traits.

## Author contributions

H.K. performed most of the experiments, with help from D.P., M.H., E.M., N.J., Y.B., and A.R.; J.M. performed LC/ESI-MS/MS and analyzed proteomics data; C.B. performed GC/MS to measure juvenile hormone levels. H.K., S.A., and A.P. designed the research, analyzed data, and wrote the manuscript. All authors read and approved a final version of this manuscript.

## Competing interests

The authors declare no competing interests.
