## [Peer Review File · Nature Communications]

REVIEWERS' COMMENTS

Reviewer #1 (Remarks to the Author):

I appreciate the strong, responsive work made by the authors. I gave them a full-court press and they did the hard work. Their new data resolve most of my issues and propels the revision toward a very good publication. Overall, this paper is among the most interesting, integrative and important reports I have seen in the field of *Drosophila* aging. It synthesizes into a coherent mechanism the processes for aging involving protein translation (think: TOR), proteostasis, reproduction, juvenile hormone (a key but usually overlooked factor), pathology, and insulin/IGF signaling. Nonetheless, I have extensive remarks for the authors. I make these to improve the work and to avoid mistakes that will diminish its impact. To keep this reasonable, below I quickly list key issues, but then attach a file with my line-by-line remarks. The authors can ignore these or make use. I suggest the latter.

Key issues:

I am still cold on the phrase “transient PT spike”. It is too contrived. A spike should be some dynamic that occurs after and before a period of homeostatic behavior or equilibrium. This is not what is observed. There is little PT at day 0, which may be expected since the flies just eclosed. It is elevated by day 2 and recedes to some quasi-stable level by age 15 days. Why not simply call the phenomenon “early-adulthood PT”?

In several places the authors engage in “Just-so-story” telling related to why (evolutionarily) PT declines. Fix this throughout least you are sent many ‘Howlers’. And ref 64 does not provide the ‘evidence’ you claim.

Your experimental design figures do not correspond to the actually collected data, and are thus misleading (eg Fig 1i,j). Revise.

The response letter said you removed the DGRP section, but the revision did not do this. Please remove, along with some other suggestions for how you can streamline and focus by taking out distracting stuff (AD, the GSC section ... which also has an experimental flaw).

JH is NOT a protein. It is a terpenoid lipid!

Details and over-the-top remarks:

I will refer by page number starting with “Main Body” and by line number within the page.

P1 line 6: To avoid misinterpretation relative to juvenile growth, try saying “We note among adults that ...”

P1 lines 8-11. Remove the sentence “Yet ... perspective.” There is no basis to speculate why early PT has not been studied. And many gero-biologists actually think that molecular events throughout life ‘drive aging’.

P1 line 16 (and throughout the paper): I am still cold on the phrase “transient PT spike”. It is still too contrived. But I don’t see that authors letting go of this term – so, readers will deliver a verdict. I say, a spike should be some dynamic that occurs after and before a period of homeostatic behavior or equilibrium. This is not what is observed. There is little PT at day 0, which may be expected since the flies have just eclosed. It is elevated by day 2 and recedes to some quasi-stable level by age 15 days. Why not simply call the phenomenon “early-adulthood PT”?

P1 lines 17-23: To avoid getting ‘howlers’ from evolutionary biologists, revise or remove these sentences. The early PT surely does participate in some physiological mechanism of antagonistic pleiotropy, but the logic stated after this will not hold up. Will the paper demonstrate ‘expense of protein homeostasis late in life’? Will it demonstrate this is caused by early gain in fecundity, and what then of how blunting early PT extends lifespan of OvoD, and works in sterile chico-homozygotes, etc? These flies lack early reproduction that would induce late costs. Are the ‘proteostatic burdens’ causal to the ‘aging process’ or correlative? I am not aware that O’Brein showed ‘evidence of selection effects as far out as day 20-30’. In fact, evolutionary biologists recognize there is very little ability for selection to operate at older ages (including in *Drosophila*) ‘to repress PT rapidly ... in order to alleviate proteostatic burden’. We don’t know why PT declines from day 2 until it reaches a steady minimum, but its level at older ages cannot be an ‘adaptive response to promote health aging.’ Given these issues, reference to the ‘protein synthesis paradox’ is irrelevant. Authors: you have a phenomenal set of designs and data. There is a gem of new biological insight in your work. I suggest that you stick close to the observed biology and retreat from selling misinterpreted concepts.

P2 line 8: Glad to see a non-feeding assay of PT, and I accept the conclusion. However, please tell readers what the assay is ... how does *in vitro* mRNA report PT ... rather than only that it obviates an artefact. The readers need to judge its validity.

P2 line 10. Operationally, you block early adult PT by treating with CHX day 0-10. This is good, but note you are not blocking a ‘spike’, which you envision to occur acutely at day 2. The semantic inconsistency in your text will confuse readers.

P3 line 9. First off, the observed shift in the fecundity schedule is stunning. This blows me away and also may hint at the underlying mechanism (diapause induced by early CHX within a sensitive window, with gradual return to reproductive syndrome). For here, rather than obliquely saying 'earlier work has found', tell the reader the specifics ... 'as seen in lines selected for survival to reproduce at late ages, early reproduction is reduced while late reproduction is enhanced'. I note as well the data overall do not suggest that elevated fecundity per se is the cause of the demographic aging. Fecundity returns to a high level at 35 days in the early CHX cohort, but PT at that time is not as high as day 2 (but is similar to day 10 when controls have peak fecundity). And then the OvoD, etc data confirm the disconnect.

Page 3 line 14: Explain design or refer to methods. As is, Fig 1 i and j seems a bit misleading: you show a hypothetical 'data plot' (i) but then only data for two of four treatments and only for the first 15 days (j). Remove fig 1i or make purely schematic (no line plot). Fix this in the other figures as well.

P4 line 18. I recommend making the subtitle more like "Induction of early-adult PT rescues longevity benefits of insulin signaling mutants"

P4 line 21. Be specific. Tell readers chico is the IRS homolog. And because you now have documented in chico heterozygotes, verbiage about 'dwarf flies' is unnecessary. Talk about IRS not dwarfs. And, the best quality chico aging data for both sexes is Tu et al (2002 Aging Cell) which is the source of your chico1 stock (the Clancy stock lacked markers, went extinct, and were not reliable). It is also important overall that chico hets have very high fecundity at young ages, while the homozygotes are basically sterile. This again disconnects PT and longevity from fecundity.

P5 line 9. These results are amazing (fig 2e). Point to Fig 2D when you say 'restored to basal level'.

P5 Line 15 etc. Yes, remove this section, as your letter mentions. DGRPs are a distraction although I do find the association very interesting. Despite your plots, it more seems to me that the distribution is binary: on or off. This suggests Mendelian segregation. It would be possible to measure PT at age 2 for the whole DGRP panel and do GWAS to identify genes and mechanisms associated with this strong variation. That could be combined with longevity data for each line, as well ... some labs have or are collecting quality life tables already. Remove this side-show for now, build a project with the full panel.

P6 line 4. You only measure P-S6K at one age after age 2. You should not say 'declines with age' based on one point (day 15) ... what if it went back up? "With age" typically means as a function of age, which means you've more than one time point.

P6 Line 6. What do you mean 'after the PT peak'? What days exactly? It seems that you started at day 2, which is 'during the peak', not after. And your data on PT only extends to day 15, not any older; we don't know what happens to PT in middle or late adulthood. It might be the

dynamics within the first 15 days alone sets up the aging pattern, and PT at later ages does not matter. Same issue when you add CHX ... you do not correct 'after early-adulthood', but during the induction window. I agree these data are really amazing but the logic does not support 'age-related decline in PT [is] ... an adaptive response to promote healthy aging.' Avoid that 'just so story' and consider: unregulated, constitutive PT (what you have induced) is bad at all ages but in newly eclosed adults, PT is transiently activated (your spike or burst) to turn on the reproductive life history mode. (You will go on to show this is more than remodeling fat body.) Semantically, on page 6 you are arguing there is an 'age-related decline in PT' but this contradicts your concept of 'transient PT spike'. A 'spike' cannot be 'age-related decline'. I keep hammering on this because you obscure fantastic data and biology with poorly constructed interpretations. Your data are simply amazing. Fix how you describe and apply logic.

P6 last paragraph. While the AD and PD data are intriguing, as with the DGRP they are too incomplete to include at this point. They will distract readers, and also open the paper for criticism from readers who specialize in AD but may not agree with your model. Save this bullet and build an independent story with fuller research in the future.

P7 last paragraph. The IUP data are interesting although their role in longevity is correlative. Whether or how such IUP affects any of the aging phenotypes ascribed to early PT is unknown: the authors must be careful viz interpretations. You apply circular logic by equating these IUP with 'proteostatic dysfunction at old age'. And you quantify ubiquitinated proteins but this is not equivalent to 'insoluble' or 'aggregated' proteins.

P8 line 5. I would say "Notably as measured in *C. elegans*, ..." And in following paragraphs segue by stating explicitly you are following up with *Drosophila*. Be specific.

Page 10, para 2. I suggest you remove the 26S discussion. It is too incomplete to make a difference and it becomes (another distraction). This is a long paper already ... focus on your story line.

P12 line 14-16. Please rewrite this sentence. It is hard to understand (and mention specifics: HSPs, *C. elegans*). The sentence is too discontinuous and vague. Try "Interestingly, proliferative GSCs in early-adulthood repress Hsps through mechanisms involving epigenetic silencing" (if I've understood you here).

Page 12 – 14. At this point of the report, the work on GSC and stress resistance, while interesting, becomes another diversion. It does not add biological insights on the pathway from PT to JH to proteostasis and aging. Yes, GSC and stress responses are a long tradition in fly/worm aging but the eyes of most readers will be glazing over by now. The section also confounds (or makes parallel) the mechanism of proteostasis, which seemed to be sufficient in terms of the many survival rescue data. To make the inference where PT > GSC > CSR > aging, you would need to block the CSR. You don't provide this test. You only have correlative data on this axis, and you've no data on 'epigenetic mechanisms'. Consider removing this section completely, work on more experiments and fire this bullet in its own publication.

P 13 last line. I do not understand what we are looking at in Fig 6f. Is this a single germarium, a developing oocyte, an egg?? There should be 1, 2 or 3 GSC per germarium. These are difficult to quantify (see papers by Niwa). I don't see the niche and I don't think you are measuring GSCs. How CHX affects GSC is NOT robustly measured.

P 14 line 2. The wording is confusing. CHX in fact confers a longevity benefit in GSC ablated flies, but the size of this benefit is less than that conferred in flies with intact GSC.

P 14 line 6. Note: OvoD retain their GSC. This is why OvoD has only a small (sometimes no) effect on survival.

P 14 line 19. 'in order to alleviate proteostatic burden' is a just-so-story. In a JSS, once you identify a function, this function becomes the explanation for why it exists ('in order to'). Avoid that trap.

P15 line 1. You might look at concepts of 'direct costs of reproduction' (O'Brein is a good resource on this). Your work demonstrates this idea, and there are very few mechanistic examples out there. You establish one mechanism of antagonistic pleiotropy, via direct COR.

P15 line 5. Strike "... and maximizing their fitness." That is confusing.

P 15 line 6. Proliferating GSC in late-adulthood? Sentence mixes up time frames.

P15 line 8 (and in your response letter): JH is **NOT** a protein!

P 15 line 9. I don't understand. What is 'generalized/bulk increase in PT', and what makes this the 'bigger picture'?

P16 second paragraph. This is a dangerous and (again) inaccurate understanding of the evolutionary dynamics of aging. It fails to understand the dynamics of selection in age-structured populations and it is a just-so-story. Having looked at reference 64 I don't understand how it showed evidence for 'selection effects as far out as day 20-30 in Drosophila.' But, this reference does discuss 'direct costs of reproduction' as a mechanism for antagonistic pleiotropy, and that is exactly what LLTPs are demonstrated to do in the current work.

Reviewer #1 (Remarks to the Author):

I appreciate the strong, responsive work made by the authors. I gave them a full-court press and they did the hard work. Their new data resolve most of my issues and propels the revision toward a very good publication. Overall, this paper is among the most interesting, integrative and important reports I have seen in the field of *Drosophila* aging. It synthesizes into a coherent mechanism the processes for aging involving protein translation (think: TOR), proteostasis, reproduction, juvenile hormone (a key but usually overlooked factor), pathology, and insulin/IGF signaling. Nonetheless, I have extensive remarks for the authors. I make these to improve the work and to avoid mistakes that will diminish its impact. To keep this reasonable, below I quickly list key issues, but then attach a file with my line-by-line remarks. The authors can ignore these or make use. I suggest the latter.

Response: We thank the reviewer for their kind words, insights, and recommendations. We have endeavored to make the requested changes including the changes sent in the reviewer's addendum. Details are provided below.

Comment 1: I am still cold on the phrase "transient PT spike". It is too contrived. A spike should be some dynamic that occurs after and before a period of homeostatic behavior or equilibrium. This is not what is observed. There is little PT at day 0, which may be expected since the flies just eclosed. It is elevated by day 2 and recedes to some quasi-stable level by age 15 days. Why not simply call the phenomenon "early-adulthood PT"?

Response: We understand the reviewer's concern. Accordingly, we have deleted the term "transient PT spike" and instead used the term "elevated early-adulthood PT". We hope that this is acceptable.

Comment 2: In several places the authors engage in "Just-so-story" telling related to why (evolutionarily) PT declines. Fix this throughout least you are sent many 'Howlers'. And ref 64 does not provide the 'evidence' you claim.

Response: This has been corrected. We have carefully revised the text to remove such 'just show' stories and ensure the manuscript describes the work precisely without drawing such inferences. We have also removed ref 64 and extensively revised the paragraph it was cited in (second to last paragraph in discussion) to be less provocative.

Comment 3: Your experimental design figures do not correspond to the actually collected data, and are thus misleading (eg Fig 1i,j). Revise.

Response: These experimental design figures were intended to help readers better understand the experimental groups for the lifespan experiments. They are color matched with the lifespans e.g. black: control, red: early-life CHX/RU486, blue: whole-life CHX/RU486, green: late-life CHX/RU486 treatment. To try to reduce confusion, we have rearranged from schematic - PT blot - lifespan to PT blot - schematic - lifespan. We hope this change makes it easier for readers to interpret.

Comment 4: The response letter said you removed the DGRP section, but the revision did not do this. Please remove, along with some other suggestions for how you can streamline and focus by taking out distracting stuff (AD, the GSC section ... which also has an experimental flaw).

Response: These have now been removed. I really apologize. These were supposed to have been removed in the prior iteration; there was a miscommunication with a member of the group when the manuscript was transferred. I have confirmed that these are now been removed.

Comment 5: JH is NOT a protein. It is a terpenoid lipid!

Response: This has been corrected.

Details and over-the-top remarks:

Comment: the reviewer provided an addendum with additional comments with the instructions. “To keep this reasonable, below I quickly list key issues, but then attach a file with my line-by-line remarks. The authors can ignore these or make use. I suggest the latter.”

Response: We deeply appreciate the reviewer taking their careful time on our manuscript. We have endeavored to make changes to accommodate all the additional comments and have tried to fully address all the ‘over the top remarks’. Details provided below

Comment 1: P1 line 6: To avoid misinterpretation relative to juvenile growth, try saying “We note among adults that ...”

Response: We corrected the sentence

Comment 2: P1 lines 8-11. Remove the sentence “Yet ... perspective.” There is no basis to speculate why early PT has not been studied. And many gero-biologists actually think that molecular events throughout life ‘drive aging’.

Response: We agree with the reviewer’s comment. This sentence has been removed.

Comment 3: P1 line 16 (and throughout the paper): I am still cold on the phrase “transient PT spike”. It is still too contrived. But I don’t see that authors letting go of this term – so, readers will deliver a verdict. I say, a spike should be some dynamic that occurs after and before a period of homeostatic behavior or equilibrium. This is not what is observed. There is little PT at day 0, which may be expected since the flies have just eclosed. It is elevated by day 2 and recedes to some quasi-stable level by age 15 days. Why not simply call the phenomenon “early-adulthood PT”?

Response: We understand the reviewer’s concern. Accordingly, we have deleted the term “transient PT spike” and instead used the term “elevated early-adulthood PT”. We hope that this is acceptable.

Comment 4: P1 lines 17-23: To avoid getting ‘howlers’ from evolutionary biologists, revise or remove these sentences. The early PT surely does participate in some physiological mechanism of antagonistic pleiotropy, but the logic stated after this will not hold up. Will the paper demonstrate ‘expense of protein homeostasis late in life’? Will it demonstrate this is caused by early gain in fecundity, and what then of how blunting early PT extends lifespan of *OvoD*, and works in sterile chichomozygotes, etc? These flies lack early reproduction that would induce late costs. Are the ‘proteostatic burdens’ causal to the ‘aging process’ or correlative? I am not aware that O’Brein showed ‘evidence of selection effects as far out as day 20-30’. In fact, evolutionary biologists recognize there is very little ability for selection to operate at older ages (including in *Drosophila*) ‘to repress PT rapidly ... in order to alleviate proteostatic burden’. We don’t know why PT declines from day 2 until it reaches a steady minimum, but its level at older ages cannot be an ‘adaptive response to promote health aging.’ Given these issues, reference to the ‘protein synthesis paradox’ is irrelevant. Authors: you have a phenomenal set of designs and data. There is a gem of new biological insight in your work. I suggest that you stick close to the observed biology and retreat from selling misinterpreted concepts.

Response: I agree. We removed the text in lines 17-23. We’ve removed the reference to the ‘protein translation paradox’. I agree that it is a bit contrived. I also agree that we can’t really ‘speak to the goals of evolution’ and have revised this paragraph to stick to the observed biology.

Comment 5: P2 line 8: Glad to see a non-feeding assay of PT, and I accept the conclusion. However, please tell readers what the assay is ... how does *in vitro* mRNA report PT ... rather than only that it obviates an artefact. The readers need to judge its validity.

Response: We added an extra sentence to better describe the assay (see below). It is also described in our methods section in more detail. I apologize for the brevity; it's a big paper with limited space. "This observation was consistent using three independent methods: 35S-methionine incorporation (Fig. 1a, left), puromycin incorporation (Fig. 1a, middle), and in vitro luciferase mRNA reporter assay¹⁷ (Fig. 1a, right). The latter assay measures the capacity of lysates to translate introduced luciferase mRNA. The luciferase assay was thus used to verify that low PT at day 0 was not an artifact of feeding labeled substrates to newly eclosed flies that may use larval acquired amino acids for PT. "

Comment 6: P2 line 10. Operationally, you block early adult PT by treating with CHX day 0-10. This is good, but note you are not blocking a 'spike', which you envision to occur acutely at day 2. The semantic inconsistency in your text will confuse readers.

Reponses: As mentioned above, we've dropped reference to a spike in favor of elevated early life PT. We also revised this sentence to "we transiently repressed PT in early-adulthood" which may be less confusing.

Comment 7: P3 line 9. First off, the observed shift in the fecundity schedule is stunning. This blows me away and also may hint at the underlying mechanism (diapause induced by early CHX within a sensitive window, with gradual return to reproductive syndrome). For here, rather than obliquely saying 'earlier work has found', tell the reader the specifics ... 'as seen in lines selected for survival to reproduce at late ages, early reproduction is reduced while late reproduction is enhanced'. I note as well the data overall do not suggest that elevated fecundity per se is the cause of the demographic aging. Fecundity returns to a high level at 35 days in the early CHX cohort, but PT at that time is not as high as day 2 (but is similar to day 10 when controls have peak fecundity). And then the OvoD, etc data confirm the disconnect.

Response: Thank you for the suggestion. We added this sentence. On the second comment, I agree we are not fully sure how to explain this either, definitely an area for further investigation!

Comment 8: Page 3 line 14: Explain design or refer to methods. As is, Fig 1 i and j seems a bit misleading: you show a hypothetical 'data plot' (i) but then only data for two of four treatments and only for the first 15 days (j). Remove fig 1i or make purely schematic (no line plot). Fix this in the other figures as well.

Response: This schematic relates to the adjacent lifespans which has all 4 groups and is color matched, red for early-life, blue for whole-life, green for late-life. To try to reduce confusion, we have rearranged from the current order schematic-PT blot-lifespan to PT blot-schematic-lifespan. I hope this change makes readers easier to interpret.

Comment 9: P4 line 18. I recommend making the subtitle more like "Induction of early-adult PT rescues longevity benefits of insulin signaling mutants"

Response: We've changed to "transiently inducing PT elevation in early-adulthood abolishes longevity benefits of insulin signaling mutants". I feel 'rescue' is potentially a confusing term as one would normally use this to refer to preventing a deficit from a knockout or disease model. 'Rescuing an increase in lifespan' I feel may confuse readers so prefer using the term 'abolish'. I hope this is acceptable.

Comment 10: P4 line 21. Be specific. Tell readers chico is the IRS homolog. And because you now have documented in chico heterozygotes, verbiage about 'dwarf flies' is unnecessary. Talk about IRS not dwarfs. And, the best quality chico aging data for both sexes is Tu et al (2002 Aging Cell) which is the source of your chico1 stock (the Clancy stock lacked markers, went extinct, and were not reliable). It is also important overall that chico hets have very high fecundity at young ages, while the homozygotes are basically sterile. This again disconnects PT and longevity from fecundity.

Response: Good suggestion. We added the below sentence and removed verbiage about 'dwarf flies'. "Mutations in *chico*, the *Drosophila* homolog of vertebrate insulin receptor substrate (IRS) have been shown to prolong life-/health-span and protect against neurodegeneration²⁵⁻²⁷." We also as suggested added reference to Tu et al.

Comment 11: P5 line 9. These results are amazing (fig 2e). Point to Fig 2D when you say 'restored to basal level'.

Response: Thank you, we have added reference to this figure.

Comment 12: P5 Line 15 etc. Yes, remove this section, as your letter mentions. DGRPs are a distraction although I do find the association very interesting. Despite your plots, it more seems to me that the distribution is binary: on or off. This suggests Mendelian segregation. It would be possible to measure PT at age 2 for the whole DGRP panel and do GWAS to identify genes and mechanisms associated with this strong variation. That could be combined with longevity data for each line, as well ... some labs have or are collecting quality life tables already. Remove this side-show for now, build a project with the full panel.

Response: I really truly apologize. The DGRP and the AD figures were supposed to have been removed in the last version. I have confirmed it has now been removed.

Comment 13: P6 line 4. You only measure P-S6K at one age after age 2. You should not say 'declines with age' based on one point (day 15) ... what if it went back up? "With age" typically means as a function of age, which means you've more than one time point.

Response: We agree. This has been revised to a more precise description of the data. "We found that T398 phosphorylation of S6K sharply declines after reaching a peak at day 2, maintaining low basal levels at day 15 (Fig. 3a)".

Comment 14: P6 Line 6. What do you mean 'after the PT peak'? What days exactly? It seems that you started at day 2, which is 'during the peak', not after. And your data on PT only extends to day 15, not any older; we don't know what happens to PT in middle or late adulthood. It might be the dynamics within the first 15 days alone sets up the aging pattern, and PT at later ages does not matter. Same issue when you add CHX ... you do not correct 'after early-adulthood', but during the induction window. I agree these data are really amazing but the logic does not support 'age-related decline in PT [is] ... an adaptive response to promote healthy aging.' Avoid that 'just so story' and consider: unregulated, constitutive PT (what you have induced) is bad at all ages but in newly eclosed adults, PT is transiently activated (your spike or burst) to turn on the reproductive life history mode. (You will go on to show this is more than remodeling fat body.) Semantically, on page 6 you are arguing there is an 'age-related decline in PT' but this contradicts your concept of 'transient PT spike'. A 'spike' cannot be 'age-related decline'. I keep hammering on this because you obscure fantastic data and biology with poorly constructed interpretations. Your data are simply amazing. Fix how you describe and apply logic.

Response: As mentioned above, we've removed reference to 'spike' in favor of elevated early adulthood PT. We measured PT in a more complete time course in Figure 1a, showing PT to either remain low or further decline from day 15 to 60. However, for this specific item, I agree that this was confusingly worded. This has been corrected to just "starting at day 2" to avoid confusion.

Comment 15: P6 last paragraph. While the AD and PD data are intriguing, as with the DGRP they are too incomplete to include at this point. They will distract readers, and also open the paper for criticism from readers who specialize in AD but may not agree with your model. Save this bullet and build an independent story with fuller research in the future.

Response: I really truly apologize. The AD and the PD figures were supposed to have been removed in the last version. I have confirmed it has now been removed.

Comment 16: P7 last paragraph. The IUP data are interesting although their role in longevity is correlative. Whether or how such IUP affects any of the aging phenotypes ascribed to early PT is unknown: the authors must be careful viz interpretations. You apply circular logic by equating these IUP with 'proteostatic dysfunction at old age'. And you quantify ubiquitinated proteins but this is not equivalent to 'insoluble' or 'aggregated' proteins.

Response: We have revised to be more circumspect in our interpretation. We have removed reference to proteostatic dysfunction instead referring specifically to impacts on IUP.

Comment 17: P8 line 5. I would say "Notably as measured in *C. elegans*, ..." And in following paragraphs segue by stating explicitly you are following up with *Drosophila*. Be specific.

Response: Agreed. we add similar text "Notably, as shown in prior *C. elegans* studies" and then specified that this represented a followup with *Drosophila* later in the paragraph.

Comment 18: Page 10, para 2. I suggest you remove the 26S discussion. It is too incomplete to make a difference and it becomes (another distraction). This is a long paper already ... focus on your story line.

Response: Agreed. We removed reference to 20S vs 26S discussion and instead just refer to proteasome

Comment 19: P12 line 14-16. Please rewrite this sentence. It is hard to understand (and mention specifics: HSPs, *C. elegans*). The sentence is too discontinuous and vague. Try "Interestingly, proliferative GSCs in early-adulthood repress Hsps through mechanisms involving epigenetic silencing" (if I've understood you here).

Response: Agree. We have modified this sentence as recommended. "Interestingly, in *C. elegans*, proliferative GSCs cause cellular stress responses (CSR) to sharply drop at the onset of reproductive maturity during early-adulthood by epigenetically silencing Hsp70 and Hsp16"

Comment 20: Page 12 – 14. At this point of the report, the work on GSC and stress resistance, while interesting, becomes another diversion. It does not add biological insights on the pathway from PT to JH to proteostasis and aging. Yes, GSC and stress responses are a long tradition in fly/worm aging but the eyes of most readers will be glazing over by now. The section also confounds (or makes parallel) the mechanism of proteostasis, which seemed to be sufficient in terms of the many survival rescue data. To make the inference where PT > GSC > CSR > aging, you would need to block the CSR. You don't provide this test. You only have correlative data on this axis, and you've no data on 'epigenetic mechanisms'. Consider removing this section completely, work on more experiments and fire this bullet in its own publication.

Response: We have tried to better clarify this section better. Our work proposes a two pathway process. Pathway 1: We report that elevated PT in early adulthood triggers JH signaling inducing elevated expression of LLTPs including vitellogenin and apolipohoron. These assist with reproduction but lead to late life proteostatic dysfunction. Pathway Two: Simultaneously we propose that, elevated early life PT enhances GSC proliferation which causes transcriptional repression of oxidative and heat stress pathways likewise contributing to reduced stress resistance and late life proteostatic dysfunction. We provide evidence that the pro-longevity effects we observe are at least in part dependent on GSC. We show that GSC ablation significantly reduces the capacity of early life PT repression to enhance lifespan. However downstream of that, I agree we cannot show that the impacts on CSR are more than correlative and have tried to make this clear in the text. I agree that this was not necessarily well articulated in the prior iteration and have added text at the start and end of this section to better explain this as well as highlight figure 6i which shows this model. We also note that reviewer 2 was quite enthusiastic about this section and a moderate portion of the work done

in this section was performed under their request. As such, we do not think we could remove this section while meeting the other reviewer's requests.

Comment 21: P 13 last line. I do not understand what we are looking at in Fig 6f. Is this a single germarium, a developing oocyte, an egg?? There should be 1, 2 or 3 GSC per germarium. These are difficult to quantify (see papers by Niwa). I don't see the niche and I don't think you are measuring GSCs. How CHX affects GSC is *NOT* robustly measured.

Response: I agree with the concerns raised. This data has been removed. It does not significantly alter our story.

Comment 22: P 14 line 2. The wording is confusing. CHX in fact confers a longevity benefit in GSC ablated flies, but the size of this benefit is less than that conferred in flies with intact GSC.

Response: I apologize. We have tried to word this in a less confusing manner based on your suggestion (see below). "In GSC-intact flies, blocking the early-life elevation in PT robustly improved lifespan (Fig. 6g; ♀: +41.5%, ♂: +42.5%). In GSC-ablated flies, early-life CHX treatment still provided a longevity benefit, but the size of this benefit is less than that conferred in GSC-intact flies (Fig. 6g; ♀: +7.9%, ♂: +9.7%)"

Comment 23: P 14 line 6. Note: OvoD retain their GSC. This is why OvoD has only a small (sometimes no) effect on survival.

Response: We agree that was our goal to use a GSC-intact but infertile line to demonstrate the impacts were not simply from impairment in fertility or reproduction per say. We have tried to clarify this better in the text (see below) "We demonstrated that blocking the early rise in PT did not improve lifespan simply by inducing sterility and abolishing survival costs of producing gametes by using a GSC-intact dominant female-sterile mutant line (*Ovo^{D1}*)."

Comment 24: P 14 line 19. 'in order to alleviate proteostatic burden' is a just-so-story. In a JSS, once you identify a function, this function becomes the explanation for why it exists ('in order to'). Avoid that trap.

Response: I agree. This has been edited to avoid such presumptions.

Comment 25: P15 line 1. You might look at concepts of 'direct costs of reproduction' (O'Brein is a good resource on this). Your work demonstrates this idea, and there are very few mechanistic examples out there. You establish one mechanism of antagonistic pleiotropy, via direct COR.

Response: We thank the reviewer for the suggestion. We think the reviewer is referring to "Use of stable isotopes to examine how dietary restriction extends *Drosophila* lifespan" this is certainly an interesting paper and an excellent example of a mechanistic investigation of antagonistic pleiotropy though this story is sufficiently distinct from ours that we were not able to find a good way to include it in the discussion. We apologize

Comment 26: P15 line 5. Strike "... and maximizing their fitness." That is confusing.

Response: Agree, it has been removed.

Comment 27: P 15 line 6. Proliferating GSC in late-adulthood? Sentence mixes up time frames.

Response: This has been reworded.

Comment 28: P15 line 8 (and in your response letter): JH is **NOT** a protein!

Response: Our apologies, this has been corrected.

Comment 29: P 15 line 9. I don't understand. What is 'generalized/bulk increase in PT', and what makes this the 'bigger picture'?

Response: We have removed this sentence.

Comment 30: P16 second paragraph. This is a dangerous and (again) inaccurate understanding of the evolutionary dynamics of aging. It fails to understand the dynamics of selection in age structured populations and it is a just-so-story. Having looked at reference 64 I don't understand how it showed evidence for 'selection effects as far out as day 20-30 in *Drosophila*." But, this reference does discuss 'direct costs of reproduction' as a mechanism for antagonistic pleiotropy, and that is exactly what LLTPs are demonstrated to do in the current work.

Response: We have removed reference 64. We have also extensively reworded p16 to be less provocative.